# MIA-DPO: Multi-Image Augmented Direct Preference Optimization For Large Vision-Language Models

**Ziyu Liu**[1,2], **Yuhang Zang**[2✉], **Xiaoyi Dong**[2], **Pan Zhang**[2], **Yuhang Cao**[2], **Haodong Duan**[2], **Conghui He**[2], **Yuanjun Xiong**[4], **Dahua Lin**[2,3,6], **Jiaqi Wang**[2,5✉]

[1] SJTU, [2] Shanghai AI Laboratory, [3]CUHK, [4] MThreads, Inc, [5] Shanghai Innovation Institute, [6] CPII under InnoHK

liuziyu77@sjtu.edu.cn, {zangyuhang, wangjiaqi}@pjlab.org.cn
Github:  https://github.com/Liuziyu77/MIA-DPO

## Abstract

Visual preference alignment involves training Large Vision-Language Models (LVLMs) to predict human preferences between visual inputs. This is typically achieved by using labeled datasets of chosen/rejected pairs and employing optimization algorithms like direct preference optimization (DPO). Existing visual alignment methods, primarily designed for single-image scenarios, struggle to effectively handle the complexity of multi-image tasks due to the scarcity of diverse training data and the high cost of annotating chosen/rejected pairs. We present **M**ulti-**I**mage **A**ugmented **D**irect **P**reference **O**ptimization (**MIA-DPO**), a visual preference alignment approach that effectively handles multi-image inputs. MIA-DPO mitigates the scarcity of diverse multi-image training data by extending single-image data with unrelated images arranged in grid collages or pic-in-pic formats, significantly reducing the costs associated with multi-image data annotations. Our observation reveals that attention values of LVLMs vary considerably across different images. We use attention values to identify and filter out rejected responses the model may have mistakenly focused on. Our attention-aware selection for constructing the chosen/rejected pairs **without** relying on (i) human annotation, (ii) extra data, and (iii) external models or APIs. MIA-DPO is compatible with various architectures and outperforms existing methods on five multi-image benchmarks, achieving an average performance boost of 3.0% on LLaVA-v1.5 and 4.3% on the recent InternLM-XC2.5. Moreover, MIA-DPO has a minimal effect on the model's ability to understand single images.

## 1 Introduction

Recent progress in Large Vision Language Models (LVLMs) marks a significant breakthrough in AI research. While proprietary models (*e.g.*, GPT-4o (OpenAI, 2024)) excel at handling multi-image contexts, current open-source LVLMs (Liu et al., 2024b;a) yield promising results but are primarily focused on *single-image* visual question answering. In real-world environments, such as digital documents and web pages, multiple figures and texts are interleaved to convey complex information effectively. The ability to understand *multi-image* contexts is a crucial direction for the future development of LVLMs.

LVLMs typically have three stages: (1) Pre-Training, (2) Supervised Fine-Tuning (SFT), and (3) Preference Alignment (*i.e.*, Reinforcement Learning from Human Feedback (RLHF) (Ouyang et al., 2022) or from AI Feedback (RLAIF) (Bai et al., 2022)). Correspondingly, to enhance the multi-image ability of LVLMs, several recent multi-image pre-training (Awadalla et al., 2023a; 2024) and instruction fine-tuning (Jiang et al., 2024; Li et al., 2024a; Chen et al., 2024b; Liu et al., 2024d) datasets and evaluation benchmarks (Jiang et al., 2024; Fu et al., 2024; Song et al., 2024; Ma et al., 2024) have been proposed. Pre-training and SFT on multi-image data can enhance the model's ability to handle multiple images to some extent. Nevertheless, similar to single-image scenarios, hallucinations remain an inevitable issue. Additionally, incorporating multi-image data during SFT

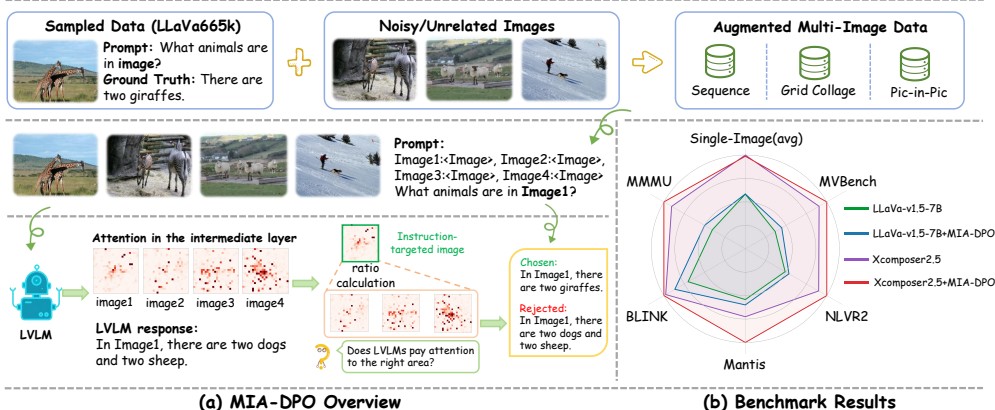

Figure 1: **(a) Overview of MIA-DPO.** We transform single-image data (*e.g.*, LLaVA 665k) into multi-image data by adding noisy or unrelated images and using language descriptions to specify the target image. Attention values are then used to detect hallucinations in multi-image contexts, filtering out rejected data for DPO optimization. **(b) Benchmark Results**. MIA-DPO excels across five multi-image benchmarks while maintaining competitive performance on seven single-image benchmarks, demonstrating its robustness in both single and multi-image tasks.

may adversely affect performance on single-image tasks. For example, previous work (Jiang et al., 2024) shows strong results on multi-image tasks after multi-image SFT but suffers a 4% average drop in single-image benchmarks. Besides pre-training and SFT, another option is preference alignment. A series of Direct Preference Optimization (DPO) (Rafailov et al., 2024) approaches in the visual domain (Sun et al., 2023; Yu et al., 2024a;b; Zhou et al., 2024) have proven effective in mitigating hallucinations in single-image scenarios. However, visual preference alignment remains little explored for multi-image fields.

Extending existing single-image preference alignment approaches to multi-image is non-trivial. A preference alignment data workflow consists of two key components: collecting question prompts, and selecting chosen/rejected response pairs. The transition to multi-image scenarios introduces the following challenges: (1) *Limited Question Prompts.* Multi-image training data is still emerging, with fewer instructions and less diversity than the extensive and varied single-image data. (2) *High Construction Costs.* Previous single-image RLHF/RLAIF approaches require high costs when constructing chosen and rejected data pairs, such as using human annotation (Sun et al., 2023; Yu et al., 2024a) or expensive GPT API (Zhao et al., 2023). Extending previous visual preference alignment data workflow to multi-image scenarios amplifies the associated costs.

To address the aforementioned challenges, we present a multi-image visual preference alignment method, dubbed as **M**ulti-**I**mage-**A**ugmented DPO (MIA-DPO). As shown in Fig. 1(a), to gather multi-image questions and answers, we extend single-image data to multi-image contexts by incorporating unrelated images, and a language description (*e.g.*, `in Image1`) to specify the target image. Additionally, we design three approaches to convert to data into a multi-image format: sequence, grid collage and pic-in-pic. This method uses existing single-image data, thereby reducing the costs associated with data collection and annotation, and is easily scalable for diverse data.

As for constructing chosen/rejected pairs, MIA-DPO eliminates the need for manual annotation or costly proprietary APIs. This is based on our observation that when LVLMs process multiple images, the attention value distribution across different images varies significantly (see bottom left of Fig. 1). We perform an **Attention Aware Selection** approach to filter out the rejected response that the attention values are mistakenly focused on irrelevant images. Our data construction method for DPO is also automated, cost-effective, and scalable for multi-image scenarios.

In summary, our key contributions are as follows:

**(1)** We first design a multi-image visual alignment pipeline MIA-DPO. Our MIA-DPO requires no manual annotations and does not rely on APIs from larger models, offering a significant cost advantage compared to existing visual alignment approaches.

**(2)** We contribute to the study of different types of multi-image hallucinations and propose to use attention values as an indicator for detecting multi-image hallucinations.

**(3)** Extensive experiments (Fig. 1(b)) demonstrate that MIA-DPO is agnostic to different LVLM architectures (LLaVA-v1.5 (Liu et al., 2024a) and InternLM-XC2.5 (Zhang et al., 2024)), boosts the performance on multiple multi-image benchmarks while maintaining the original single-image understanding capabilities.

## 2    RELATED WORKS

**Large Vision Language Models** (LVLMs), like GPT-4V (Achiam et al., 2023), signify a major breakthrough in the development of Large Language Models (LLMs) by incorporating both visual and textual data (Bai et al., 2023; Wang et al., 2024a;b). LVLMs significantly enhance the quality of human-AI interactions, making these exchanges more intuitive and seamless. To enable LVLMs to handle multi-image tasks, several multi-image datasets suitable for pre-training and supervised fine-tuning (SFT) have gradually emerged (Jiang et al., 2024; Liu et al., 2024d; Song et al., 2024). However, due to the lag in the development of multi-image datasets, data and methods tailored for multi-image tasks during the RLHF/RLAIF phase remain unexplored. Therefore, we designed a dedicated MIA-DPO framework for multi-image tasks, aimed at improving the ability of LVLMs to handle multi-image scenarios.

**Visual Preference Alignment** is a multi-modal extension of preference alignment techniques with image inputs. Preference alignment aligns LLMs with human values and reduces hallucinations by collecting pairs of preferred and rejected data, using optimization techniques including PPO (Schulman et al., 2017) and DPO (Rafailov et al., 2024) to guide the model's adjustments. Earlier approaches, such as LLaVa-RLHF (Sun et al., 2023) and RLHF-V (Yu et al., 2024a), required human labeling of preferred data, which incurs high labor costs. HA-DPO (Zhao et al., 2023) mitigates this by using GPT-4's API to generate the necessary DPO data, but it still faces high API costs. RLAIF-V (Yu et al., 2024b) employs a text-splitting approach to scoring individual text segments and using open-source LVLMs for data generation. POVID (Zhou et al., 2024) uses blurred images and GPT-4 to inject hallucinations to construct the DPO data. The previous approaches focus solely on single-image scenarios and require costly chosen/rejected data. Our MIA-DPO first enables visual preference alignment for multi-image scenarios and achieves low-cost DPO data construction.

## 3    METHODS

We first introduce the background of visual preference alignment in Sec. 3.1. We analyze the multi-image hallucinations in Sec. 3.2. We present our MIA-DPO framework in Sec. 3.3.

### 3.1    PRELIMINARY

To enhance LVLMs' understanding of multi-image inputs, we employ visual preference alignment. This section introduces the concept of visual preference alignment and highlights the DPO approach as a representative example.

**Visual Preference Alignment**    Preference alignment aims to align a model's preferences with human preferences. Representative approaches include **R**einforcement **L**earning from **H**uman **F**eedback (**RLHF**) (Ouyang et al., 2022) and **R**einforcement **L**earning from **AI** **F**eedback (**RLAIF**) (Bai et al., 2022). Given a dataset $D$[1], where each sample consists of an input prompt $x$, the chosen answers $y_w$ and the rejected output $y_l$. We can represent $D$ as follows: $D = \{x, y_w, y_l\}$. The input prompt $x$ can be an interleaved sequence of images $v$ and texts $t$. When an LVLM processes an input $x$ and generates an output $y$, a reward $r(x, y)$ is assigned. The reward model $r$ assesses the chosen (high value of $r(x, y)$) and rejected (low $r(x, y)$) samples. Visual preference alignment aims to **maximize** the reward $r(x, y)$:

$$\max_{\theta} \mathbb{E}_{x \sim D, y \sim \pi_\theta(y|x)} \left[ r(x, y) \right], \tag{1}$$

[1]For simplicity, we use a single sample in our formulations, which can be easily extended to a batch of samples.

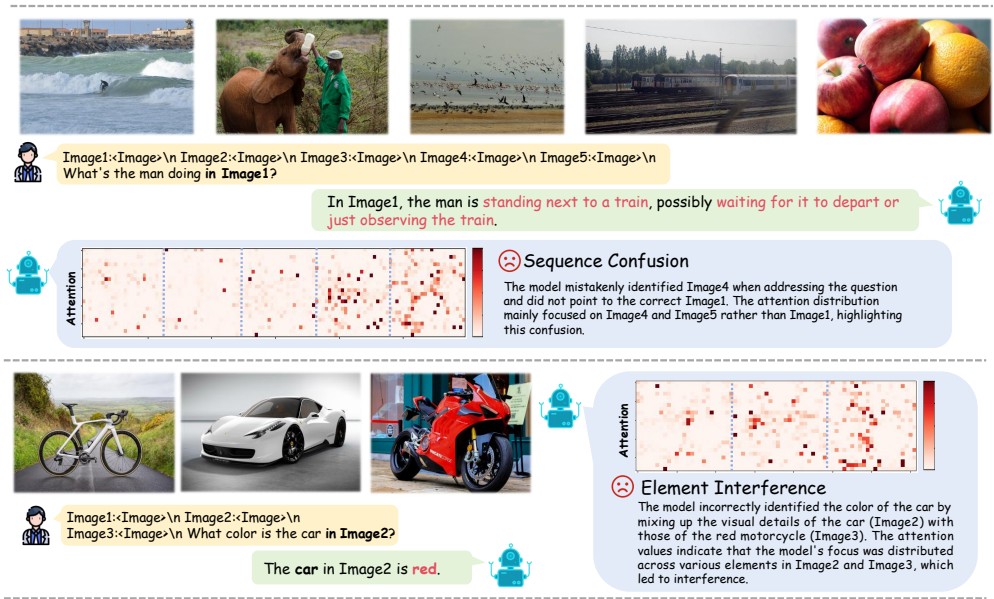

Figure 2: **Examples of Multi-Image Hallucinations.** **Top**: *Sequence Confusion* that the model is confused about the order in which the images should be referenced. **Bottom**: *Element Interference*. The model incorrectly identified the attributes due to visual element interference across different images. **Attention values** illustrate how the model's focus was dispersed across different images, resulting in the hallucination response.

while $\theta$, $\pi_\theta$ and $\pi_\theta(y|x)$ refer to the parameter, policy, and output distribution of LVLM, respectively.

To prevent over-fitting to the dataset $D$, preference alignment approaches incorporate a KL-divergence loss $D_{\text{KL}}$ to regularize the difference between the model's policy $\pi_\theta(y|x)$ and a reference model's policy $\pi_{\text{ref}}(y|x)$:

$$\max_\theta \left[ \mathbb{E}_{x \sim D, y \sim \pi_\theta(y|x)} \left[ r(x, y) \right] - \beta \cdot D_{\text{KL}}(\pi_\theta(y|x) \parallel \pi_{\text{ref}}(y|x)) \right], \tag{2}$$

where the hyper-parameter $\beta$ controls the influence of KL-divergence on the optimization objective. The reference model is the model's state prior to preference alignment.

**Direct Preference Optimization (DPO)**   To optimize the preference alignment objective in Eq. (2), we can use either an online reward model (*e.g.*, PPO (Schulman et al., 2017)) or precomputed off-line chosen/rejected pairs (*e.g.*, DPO (Rafailov et al., 2024)). Given its simplicity, DPO has been widely adopted in previous visual alignment works (Sun et al., 2023; Yu et al., 2024a; Zhao et al., 2023; Yu et al., 2024b). We reformulate Eq. (2) as the loss function of DPO:

$$\mathcal{L}_{\text{DPO}}(\pi_\theta; \pi_{\text{ref}}) = -\mathbb{E}_{(x, y_w, y_l) \sim \mathcal{D}} \left[ \log \sigma \left( \beta \log \frac{\pi_\theta(y_w|x)}{\pi_{\text{ref}}(y_w|x)} - \beta \log \frac{\pi_\theta(y_l|x)}{\pi_{\text{ref}}(y_l|x)} \right) \right], \tag{3}$$

where $\sigma(.)$ denotes the sigmoid function. As shown in Eq. (3), DPO-based alignment methods focus on constructing input prompts $x$ (see Sec. 3.3.1), and selecting chosen $y_w$ and rejected $y_l$ pairs (see Sec. 3.3.2).

## 3.2   ANALYSIS ON MULTI-IMAGE HALLUCINATIONS

In this section, we conduct various studies to analyze the characteristics of multi-image hallucinations in LVLMs and reveal that the attention mechanism is a proper indicator to determine when hallucinations occur.

**Two-types of Multi-Image Hallucinations**   Some previous studies (Li et al., 2023c; Ouali et al., 2024) have explored different types of single-image hallucinations, such as object hallucination which means the model incorrectly describes objects that are not present in the image. Compared to

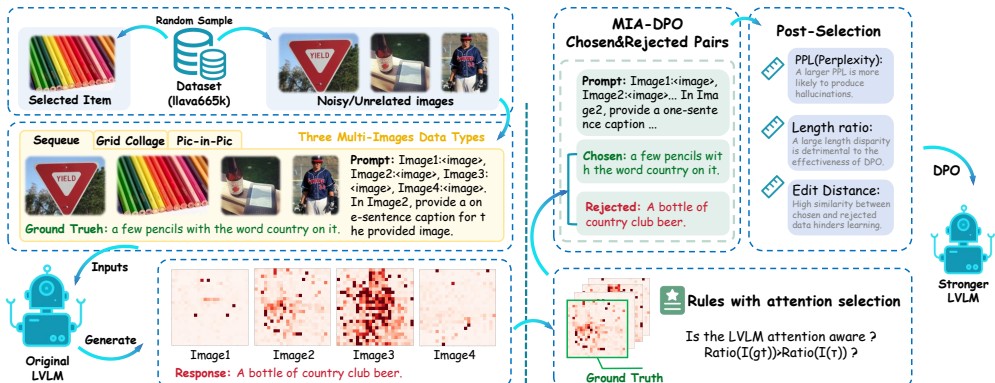

Figure 3: **MIA-DPO Framework**. We extend the single-image dataset to multi-image datasets by inserting irrelevant images and using attention values to filter out the hallucination responses for rejected samples of the DPO algorithm.

single-image hallucinations, multi-image scenarios introduce more complex types of hallucinations. As shown in Fig. 2, we categorize multi-image hallucinations into two-types:

(1) *Sequence Confusion*. When presented with multiple images, the model may fail to identify which image the input prompt refers to. For instance, in the top case shown in Fig. 2, the question is directed at Image 3 (birds and sky), but the model responds based on Image 4 (a train on tracks).

(2) *Element Interference*. The presence of multiple images significantly increases the number of visual elements compared to a single image, leading to confusion between different elements by LVLMs. For example, in the bottom case of Fig. 2, the question "What color is the car in Image2?" should be answered with "white". However, the LVLM incorrectly interpreted the color attribute of the motorcycle in Image 3 as the color of the car in Image 2, resulting in an incorrect response.

**Attention as an Indicator for Detecting Hallucinations** The attention mechanism reveals where the model is "looking" when making a decision. We observe that the attention mechanism provides crucial clues for detecting multi-image hallucinations (Fig. 2). Ideally, attention values should focus on areas of the referred input image relevant to the question. If the attention values are scattered or not strongly focused on the correct visual element or region, it suggests the model is experiencing difficulty understanding multi-image sequences or distinguishing elements between different images. Based on our observation, we design an attention-aware selection that uses the attention values to select the rejected sample that contains the hallucinations in the DPO algorithm (Sec. 3.3.2).

## 3.3 MIA-DPO Framework

As illustrated in Fig. 3, MIA-DPO initially extends single-image prompts to multi-image prompts (Sec. 3.3.1), followed by attention-based filtering of rejected data and post-selection processing (Sec. 3.3.2). Finally, we apply the DPO algorithm (Sec. 3.3.3) to the constructed multi-image prompts and chosen/rejected pairs, resulting in a stronger model.

### 3.3.1 From Single-Image Prompts to Multi-Image Prompts

Rather than expending effort on collecting and annotating new multi-image prompts, we efficiently convert existing single-image datasets, such as LLaVA-665k (Liu et al., 2024a), by incorporating unrelated images. Our low-cost, scalable approach enriches data forms and allows us to comprehensively explore the various types of multi-image hallucinations that LVLMs might produce.

As shown in Fig. 4, we construct multi-image prompts in three formats: (1) **Sequence:** Multiple images are arranged sequentially, with questions targeting specific images. The number of images varies from 2 to 5. (2) **Grid Collage:** Multiple images are merged into a single image, each labeled with a number description. Questions focus on specific images based on language descriptions. The number of images ranges from 2 to 9. (3) **Pic-in-Pic:** One image is resized and overlaid onto another, and questions are asked about the combined image.

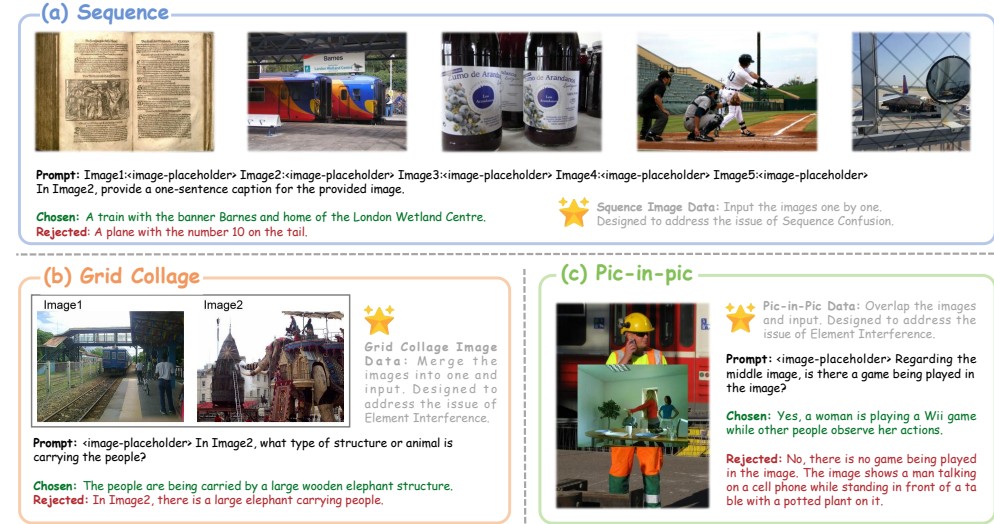

Figure 4: **Multi-Images DPO Data Format.** To address multi-image hallucinations mentioned in Fig. 2, we construct our multi-image prompts in three formats: (a) Sequence. (b) Grid Collage. (c) Pic-in-Pic.

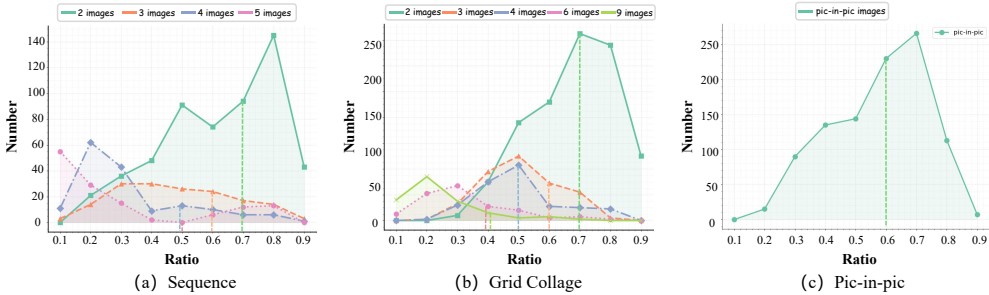

(a) Sequence                (b) Grid Collage                (c) Pic-in-pic

Figure 5: **Attention Ratio Statistic.** We analyze the attention ratios distribution for different image counts across various data types, and use dashed lines to indicate the thresholds for each data set.

These three data types are specifically designed to address the two types of multiple-image hallucinations in Fig. 2. *Sequence* data extends the overall length of image tokens and introduces multiple unrelated images to confuse the LVLMs, challenging their ability to determine image order (Sequence Confusion in Fig. 2). *Grid Collage* and *Pic-in-Pic* data stack multiple images, increasing the likelihood of LVLMs confusing image elements and failing to accurately locate the content based on language descriptions (Element Interference in Fig. 2). Our diverse multi-image prompts enhance data richness, address various types of multi-image hallucinations, and provide a strong foundation for constructing chosen/rejected pairs.

### 3.3.2 ATTENTION-AWARE SELECTION FOR REJECTED SAMPLES

As we analyzed in Sec. 3.2, the model's attention values are clues for detecting multi-image hallucinations. Inspired by our observation, we present an attention-aware selection mechanism for constructing the rejected samples of the DPO algorithm.

Given the input question $x$ and a set of generated answers $(y_1, y_2, \ldots) \sim \pi_\theta(y|x)$. For each answer sample $y$, we compute the attention value metric $R(y) = \frac{A_{\text{target}}}{A_{\text{sum}}}$, where $A_{\text{target}}$ be the amount of attention directed toward the target defined in $x$, and $A_{\text{total}}$ be the total amount of attention values. By setting an attention ratio threshold $\tau$, we can select cases $y_l$ that the LVLMs did not correctly focus on the image or region specified:

$$y_l = \{y \mid y \sim \pi_\theta(y|x) \text{ and } R(y) \leq \tau\}. \tag{4}$$

We use $y_l$ as the rejected answer for the DPO algorithm. We use the ground truth of question $x$ as the chosen sample $y_w$. Finally, we construct the DPO pair data $D = \{x, y_w, y_l\}$ in Eq. (3).

**Determining the Ratio Threshold**    The sequence data include sets of 2, 3, 4, and 5 images. For each set, we calculated the proportion of attention focused on the image relevant to the question or instruction, relative to the total attention across all input images. Our statistical results are visualized in Figure 5(a). As shown in Fig. 5(a), the average attention ratio decreases as the number of images increases, though the overall distribution trend remains consistent. Based on our findings, we set attention ratio thresholds at 0.7, 0.6, 0.5, and 0.5 for sets of 2, 3, 4, and 5 images, respectively. Data below the thresholds is marked as rejected. We applied the same statistical approach for grid collage data and pic-in-pic data, and visualized the results in Fig. 5(b) and 5(c). For grid collage image data with 2, 3, 4, 6, and 9 images, we set the value of $\tau$ as 0.7, 0.6, 0.5, 0.4, and 0.4, respectively. For pic-in-pic data, we set $\tau = 0.6$.

**Post-Selection for Data Cleaning**    Although our attention-aware selection is effective in constructing the DPO data, a small amount of noisy samples may be included and potentially causing detrimental effects. To filter out the noisy samples, we incorporate a post-selection step using the following three metrics: (1) **Perplexity (PPL)**. The PPL metric measures the negative log-likelihood of the generated sequence, and is a common metric for data cleaning (Albalak et al., 2024). A high PPL value suggests that LVLMs have lower confidence and are more likely to contain hallucinations. We use the PPL metric to filter out low confidence responses. (2) **Length Ratio**. Previous studies (Singhal et al., 2023; Dubois et al., 2024) have shown that the reward model may favor lengthier content. To mitigate the length bias, we compute the length difference between the chosen and rejected data, excluding the samples where the difference value is too large. (3) **Edit Distance**. We observed that some samples may not contribute meaningfully to the optimization process. For example, the difference between "apple" (chosen) and "apples" (rejected) is minimal in terms of edit distance, which is less useful for distinguishing patterns. We use the edit distance to ensure the DPO process does not incorporate pairs with excessively small differences.

The post-selection approach will filter out approximately 5% of the data. We provide the ablation studies in Sec. 4.4 to demonstrate that our post-selection helps maintain the high quality of data.

### 3.3.3    OPTIMIZATION

As discussed in Sec. 3.3.1 and Sec. 3.3.2, we have outlined how to construct multi-image input prompts $x$, and select chosen $y_w$ and rejected $y_l$ pairs. By applying Eq. (3), we can update the policy $\pi_\theta$.

To improve the stability of DPO training, following the approach in (Dubey et al., 2024; Pang et al., 2024), we add a negative log-likelihood(NLL) loss $\mathcal{L}_{\mathrm{NLL}}(\pi_\theta) = -\log \pi_\theta(y_w|x)$. We use a parameter $\gamma$ to balance the $\mathcal{L}_{\mathrm{DPO}}$ and $\mathcal{L}_{\mathrm{NLL}}$. The final loss $\mathcal{L}_{\mathrm{total}}$ is defined in Eq. (5):

$$\mathcal{L}_{\mathrm{total}} = \mathcal{L}_{\mathrm{DPO}}(\pi_\theta; \pi_{\mathrm{ref}}) + \gamma \mathcal{L}_{\mathrm{NLL}}(\pi_\theta). \tag{5}$$

## 4    EXPERIMENTS

### 4.1    EXPERIMENTAL SETUP

**Benchmarks**    We evaluate our method on the following representative benchmarks. First, we select five **multi-image** benchmarks: MMMU (Yue et al., 2024), BLINK (Fu et al., 2024), Mantis (Jiang et al., 2024), NLVR2 (Suhr et al., 2018), and MVBench (Li et al., 2024c). The MMMU benchmark includes questions involving both single-image and multi-image scenarios. Subsequently, we also test the model on several **single-image** benchmarks: MMStar (Chen et al., 2024a), ScienceQA (Lu et al., 2022), MMVet (Yu et al., 2023), POPE (Li et al., 2023c), MMBench (Liu et al., 2023), MathVista (Lu et al., 2023), AI2D (Kembhavi et al., 2016), and OCRBench (Liu et al., 2024c). We evaluate our method on a diverse set of benchmarks, demonstrating its effectiveness across both scenarios. These evaluations confirm the model's improved performance, particularly in multi-image contexts.

**Baseline Methods**    We compare MIA-DPO with three preference optimization baselines. (1) LLaVA-RLHF (Sun et al., 2023) improves model performance by augmenting GPT-4-generated data with existing human-written image-text data.(2) HA-DPO (Zhao et al., 2023) uses GPT-4 to detect and correct hallucinations in the model's responses. (3) POVID (Zhou et al., 2024) prompts GPT-4V

Table 1: **Main results on multi-image benchmarks.** We compare our MIA-DPO along with other DPO algorithms across five multi-image benchmarks. Our method brings significant performance improvements to both the classic LLaVa-v1.5 and the recent InternLM-XC2.5. In contrast, other single-image DPO methods perform poorly on multi-image benchmarks.

| Models | Parameter | MMMU | BLINK | Mantis | NLVR2 | MVBench | Average |
|---|---|---|---|---|---|---|---|
| GPT-4V (Achiam et al., 2023) | - | 56.8 | 51.1 | 62.7 | 88.8 | 43.5 | 60.6 |
| LLaVA-v1.6 (Li et al., 2024b) | 7B | 35.8 | 39.6 | 45.6 | 58.9 | 40.9 | 44.2 |
| Qwen-VL-Chat (Bai et al., 2023) | 7B | 35.9 | 31.2 | 39.2 | 58.7 | 42.2 | 41.4 |
| VideoLLaVA (Lin et al., 2023) | 7B | - | 38.9 | 35.9 | 56.5 | 44.3 | - |
| Fuyu (Bavishi et al., 2023) | 8B | 27.9 | 36.6 | 27.2 | 51.1 | 30.2 | 34.6 |
| Idefics2 (Laurençon et al., 2024b) | 8B | 43.0 | 45.2 | 48.9 | 86.9 | 29.7 | 50.7 |
| InstructBLIP (Dai et al., 2023) | 13B | 30.6 | 42.2 | 45.6 | 60.3 | 32.5 | 42.2 |
| CogVLM (Wang et al., 2023) | 17B | 32.1 | 41.5 | 45.2 | 58.6 | 37.3 | 42.9 |
| Emu2-Chat (Sun et al., 2024) | 37B | 36.3 | 36.2 | 37.8 | 58.2 | 39.7 | 41.6 |
| LLaVA-v1.5 (Liu et al., 2024a) | 7B | 35.1 | 37.1 | 41.9 | 52.1 | 36.0 | 40.4 |
| + LLaVA-RLHF (Sun et al., 2023) | 7B | 34.6 | 40.8 | 30.4 | 51.8 | 38.0 | 39.1 |
| + HA-DPO (Zhao et al., 2023) | 7B | 35.8 | 38.6 | 34.6 | 51.6 | **40.6** | 40.2 |
| + POVID (Zhou et al., 2024) | 7B | 35.2 | 19.9 | 37.8 | 21.4 | 39.4 | 30.7 |
| + MIA-DPO (Ours) | 7B | **36.3** | **42.9** | **44.2** | **54.2** | 39.5 | **43.4** |
| Δ | - | +1.2 | +5.8 | +2.3 | +2.1 | +3.5 | +3.0 |
| InternLM-XC2.5 (Zhang et al., 2024) | 7B | 41.4 | 46.9 | 49.3 | 70.7 | 59.5 | 53.6 |
| + HA-DPO (Zhao et al., 2023) | 7B | 42.0 | 46.9 | 51.6 | 71.6 | 58.0 | 54.0 |
| + POVID (Zhou et al., 2024) | 7B | 42.4 | 47.9 | 51.2 | 70.6 | 59.2 | 54.3 |
| + MIA-DPO (Ours) | 7B | **42.6** | **47.7** | **60.4** | **75.2** | **63.6** | **57.9** |
| Δ | - | +1.2 | +0.8 | 11.1 | +4.5 | 4.1 | +4.3 |

to inject plausible hallucinations into correct answers, followed by image distortion to provoke the LVLMs' inherent tendency towards hallucinations.

**Implementation Details** Our MIA-DPO is applicable to various LVLMs. We select two models in our experiments: the classic LLaVA-v1.5 (Liu et al., 2024a) and the recent InternLM-XC2.5 (Zhang et al., 2024). The models are trained on 3 epochs, with a learning rate of $5e-5$, temperature parameter (in Eq. 3) $\beta = 0.1$, and NLL loss coefficient (in Eq. 5) $\gamma = 0.1$. For more experimental details, please refer to appendix Sec. A.

## 4.2 RESULTS ON MULTI-IMAGES BENCHMARKS

**Results on LLaVA-v1.5** As present in Tab. 1, applying MIA-DPO to LLaVA-v1.5 achieves improvements of $1.2\%/5.8\%/2.3\%/2.1\%/3.5\%$ on five multi-image benchmarks, which demonstrates the effectiveness of MIA-DPO. As for the challenging MMMU benchmark that requires complex domain-specific knowledge, MIA-DPO enables LLaVA-v1.5 to achieve a $1.2\%$ improvement. The experimental results on MMMU demonstrate that MIA-DPO enhances the LLaVA-v1.5's reasoning ability on multi-image problems. Additionally, on the BLINK dataset that includes multi-view and spatial relationship reasoning, MIA-DPO significantly boosts the performance of LLaVA-v1.5 by $5.8\%$. Such an improvement highlights the effectiveness of MIA-DPO in enhancing the model's ability to understand and reason under multi-image scenarios.

**Comparison with Preference Optimization Baselines** In Tab. 1, we compare MIA-DPO with three preference optimization baselines (LLaVA-RLHF, HA-DPO, POVID) on LLaVA-v1.5. Thanks to our multi-image attention-based method for constructing the DPO data, MIA-DPO achieves significant advantages on the reported five multi-image benchmarks compared to the baselines.

**More LVLM Architectures** We also applied MIA-DPO to other LVLM architectures, such as the recent InternLM-XC2.5 model. As shown in Tab. 1, MIA-DPO boosts the performance of $1.2\%/0.8\%/11.1\%/4.5\%/4.1\%$ across the five benchmarks, resulting in an average improvement of $4.3\%$. The results on LLaVA-1.5 and InternLM-XC2.5 demonstrate that MIA-DPO is general and effective for different LVLM architectures. Notably, despite the Supervised Fine-tuning (SFT) phase of InternLM-XC2.5 involving multi-image data, our MIA-DPO still further boosts performance on multi-image benchmarks.

Table 2: **Main results on single-image benchmarks.** We compare MIA-DPO with other DPO approaches across seven single-image benchmarks. MIA-DPO, which not only enhances multi-image performance but also maintains strong proficiency in single-image tasks.

| Models | Parameter | MMStar | SQA | MMVet | POPE | MMB | Math | AI2D | OCR | Average |
|---|---|---|---|---|---|---|---|---|---|---|
| LLaVA-v1.6 (Li et al., 2024b) | 7B | 37.6 | 87.5 | 40.2 | 70.3 | 69.8 | 31.5 | 67.0 | 53.7 | 57.2 |
| Qwen-VL-Chat (Bai et al., 2023) | 7B | 34.5 | 68.8 | 47.3 | 74.9 | 61.8 | 15.5 | 63.0 | 48.8 | 51.8 |
| Idefics2 (Laurençon et al., 2024b) | 8B | 49.5 | 88.7 | 34.0 | 86.2 | 75.7 | 51.4 | 72.3 | - | - |
| OpenFlamingo (Awadalla et al., 2023b) | 9B | 36.9 | 44.8 | 23.2 | 52.6 | 32.4 | 18.6 | 31.7 | 14.9 | 31.9 |
| InstructBLIP (Dai et al., 2023) | 13B | 32.7 | 54.1 | 33.1 | 86.1 | 38.3 | 24.4 | 40.6 | 27.6 | 42.1 |
| CogVLM (Wang et al., 2023) | 17B | 39.9 | 66.2 | 54.5 | 88.0 | 65.8 | 35.0 | 63.3 | 59.0 | 59.0 |
| Emu2-Chat (Sun et al., 2024) | 37B | 40.7 | 68.2 | 31.0 | 88.0 | 63.4 | 30.7 | 49.7 | 43.6 | 51.9 |
| LLaVA-v1.5 (Liu et al., 2024a) | 7B | 32.9 | 66.6 | 30.5 | 85.9 | 64.3 | 25.4 | 55.5 | 31.8 | 49.1 |
| + LLaVA-RLHF Sun et al. (2023) | 7B | 31.6 | 64.0 | 27.8 | 80.8 | 60.1 | 23.5 | 47.9 | 28.0 | 45.5 |
| + HA-DPO (Zhao et al., 2023) | 7B | 33.5 | 67.3 | 29.1 | 84.3 | 64.9 | 25.8 | 53.9 | 31.1 | 48.7 |
| + POVID (Zhou et al., 2024) | 7B | 36.2 | 68.8 | 31.8 | 86.3 | 64.9 | 24.4 | 55.2 | 31.6 | 49.9 |
| + MIA-DPO (ours) | 7B | 32.9 | 67.6 | 32.1 | 87.2 | 63.1 | 24.4 | 54.7 | 30.5 | 49.1 |
| InternLM-XC2.5 (Zhang et al., 2024) | 7B | 59.7 | 96.3 | 48.7 | 87.9 | 81.9 | 63.3 | 81.5 | 69.0 | 73.5 |
| + HA-DPO (Zhao et al., 2023) | 7B | 59.6 | 96.2 | 53.3 | 84.4 | 81.4 | 62.6 | 81.8 | 68.2 | 73.4 |
| + POVID (Zhou et al., 2024) | 7B | 59.7 | 96.2 | 54.8 | 88.1 | 81.4 | 62.6 | 81.3 | 68.9 | 74.1 |
| + MIA-DPO (ours) | 7B | 61.1 | 96.2 | 46.7 | 86.9 | 80.4 | 61.7 | 81.6 | 67.4 | 72.8 |

## 4.3 RESULTS ON SINGLE-IMAGES BENCHMARKS

While MIA-DPO is effective in multi-image scenarios, we also report the performance on single-image benchmarks. As shown in Tab. 2, MIA-DPO outperforms the LLaVA-v1.5 baseline and DPO methods, including LLaVA-RLHF and HA-DPO, in average results across seven single-image benchmarks. As for the InternLM-XC2.5 model, MIA-DPO achieves a $1.4\%$ increase on MMStar but performs slightly below baseline on average across all single-image benchmarks. The slight degradation in InternLM-XC2.5's single-image performance suggests that while the model benefits greatly in multi-image scenarios, there may be a trade-off in optimizing for more complex, inter-leaved inputs. Overall, our findings highlight the robustness of our MIA-DPO, which not only excels in improving multi-image performance but also preserves proficiency on single-image tasks. Our MIA-DPO serves as a strong candidate for real-world applications requiring versatile multi-modal abilities across both single and multiple image tasks.

## 4.4 ABLATION STUDIES

**Ablation Studies on Post-Selection** In our ablation study, we experimented with the post-selection process for DPO data. As illustrated in Fig. 3, our post-selection process includes three components: perplexity (ppl), text length, and edit distance. We conduct ablation studies to compare the impact of whether to use the post-selection or not. In Tab. 3, the results show that while MIA-DPO without post-selection (row 1) still led to improvements across multiple multi-image benchmarks, its performance was consistently lower than that of MIA-DPO with post-selection (row 2). Our findings highlight that post-selection effectively removes outlier and low-quality data, further enhancing the overall quality of the DPO pair data and boosting model performance.

**Ablation Studies on Data Types** In the process of constructing multi-image DPO data for MIA-DPO, we created three types of data: Sequence, Grid Collage, and Pic-in-Pic Data. These three types of data work together to specifically eliminate the two types of multi-image hallucinations we identified: Sequence Confusion and Element Interference. To study the impact of each data type on overall performance, we trained the LLaVa-v1.5 model separately with 20k instances of each data type and summarized the results in Tab. 3.

The experimental results indicate that using each data type individually for DPO on LLaVa-v1.5 yields similar average scores of 42.6, 42.4, and 42.7 across five benchmarks. However, when combining all three data types, the model achieves a higher average score of 43.4, as shown in Tab. 1. This suggests that the three data types address different hallucination types, and their combination produces better results than using them separately.

| # | | MMMU | BLINK | Mantis | NLVR2 | MVBench | Average |
|---|---|---|---|---|---|---|---|
| | | 35.1 | 37.1 | 41.9 | 52.1 | 36.0 | 40.4 |
| 1 | w/o post sel. | 35.3 | 38.7 | 44.2 | 53.7 | 39.4 | 42.3 |
| 2 | w post sel. | 36.3 | 42.9 | 44.2 | 54.2 | 39.5 | **43.4** |
| 3 | sequence | 37.3 | 39.5 | 44.2 | 51.7 | 40.1 | 42.6 |
| 4 | grid collage | 37.1 | 40.4 | 44.2 | 51.0 | 39.4 | 42.4 |
| 5 | pic-in-pic | 37.9 | 40.8 | 41.9 | 53.2 | 39.8 | 42.7 |

Table 3: **Ablation Studies.** The top row refers to the LLaVA-v1.5 baseline. We conduct experiments about the impact of without (w/o) and with (w) post-selection techniques and dpo data types.

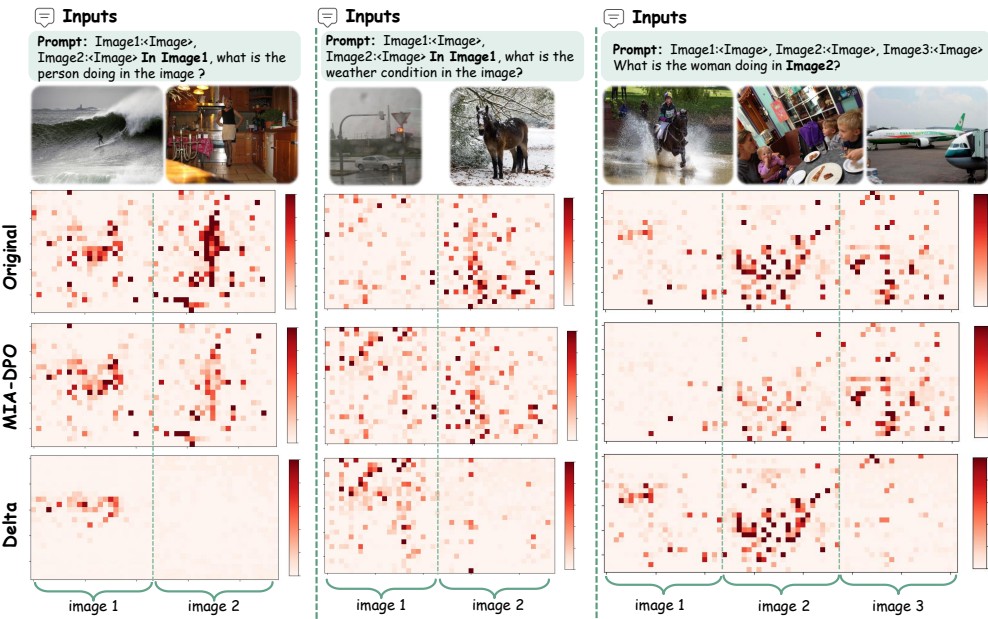

Figure 6: **Attention Difference Before and After DPO.** We present the attention distribution in the intermediate layers for the original LLaVA-v1.5 (**top row**), MIA-DPO + LLaVA-v1.5 (**second row**), and the difference value (**bottom row**), respectively.

## 4.5 VISUALIZATION OBSERVATIONS

We visualize the reasoning process of the LLaVA-v1.5 model before and after applying MIA-DPO on multi-image cases. In Fig. 6, we show the attention map of the generated text tokens relative to the input image tokens. The top and second rows display the attention distribution before and after applying MIA-DPO, respectively. The attention difference (delta value) in the third row indicates which areas receive increased attention due to applying our preference optimization process.

Using MIA-DPO, the LLaVA-v1.5 model adjusts its focus to specific image regions corresponding to the given instruction. In both the first and second cases, we observe an increased focus on the instruction-targeted areas of Image 1 after applying MIA-DPO. In the third case, attention gravitates more toward Image 2, which is specified in the language instruction. The visualization results indicate that MIA-DPO effectively improves the model's ability to correctly allocate attention to the relevant image regions, reducing the likelihood of multi-image hallucinations.

## 5 CONCLUSION

Aligning models with human preferences is a critical goal. In this paper, we are the first to propose a multi-image DPO framework. We conducted an in-depth analysis of the differences between hallucinations in multi-image and single-image reasoning for LVLMs, exploring the root causes of multi-image hallucinations through the lens of attention. Our findings reveal that a lack of attention-aware capabilities is a key factor contributing to hallucinations in multi-image reasoning. Based on these insights, we introduced MIA-DPO (Multi-Image Augmented Direct Preference Optimization). Results from tests on five multi-image benchmarks and seven single-image benchmarks demonstrate that MIA-DPO significantly improves the model's performance in multi-image reasoning while maintaining its original single-image reasoning capabilities.

**Acknowledgments**

This project is funded in part by Shanghai Artificial lntelligence Laboratory, Shanghai Innovation Institute, the National Key R&D Program of China (2022ZD0160201), the Centre for Perceptual and Interactive Intelligence (CPII) Ltd under the Innovation and Technology Commission (ITC)'s InnoHK. Dahua Lin is a PI of CPII under the InnoHK.

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

APPENDIX

In this appendix, we provide additional supporting materials to facilitate a deeper understanding of our work. First, in Sec. A, we further enrich the experiments, including ablation studies and experimental details. In Sec. B, we list all the models and benchmarks we used, along with a statistical overview of the amount and ratio of data utilized in MIA-DPO. In Sec. C, we present more examples of the three data types: Sequence Data, Grid Collage Data, and Pic-in-Pic Data. In Sec. D, we share our observations on the attention distribution in LVLMs multi-image reasoning, explaining the basis for attention-aware selection.

# A  MORE EXPERIMENTS

## A.1  ABLATION STUDIES

**Ablation Studies on $\gamma$ and Epochs**   We perform ablation studies on the key hyper-parameters, including the NLL loss coefficient $\gamma$ and the number of training epochs. As shown in Tab. 4, we observe that a larger value of $\gamma$ negatively impacts the training process, while the number of epochs has a minor effect on the final results. Based on the experimental results, we set 3 epochs and $\gamma = 0.1$ as the default values for the parameters.

**GPT-4o-mini Selection and MIA-DPO**   To validate the effectiveness of MIA-DPO, we introduce an ablation experiment using GPT-4o-mini for DPO data selection. The process begins with the model generating answers to our multi-image questions, followed by presenting both the model's responses and the ground truth to GPT-4o-mini. GPT-4o-mini then assesses the accuracy of the model's responses and their similarity to the ground truth, assigning a score between 0 and 10 based on various criteria. We classify responses with scores below 7 as rejected data and use them to construct the DPO data. The results are presented in Tab. 5. Our observations indicate that MIA-DPO not only offers a cost advantage over the GPT-4o-mini-based data selection method but also outperforms it across five benchmarks.

The prompt we use to guide GPT-4o-mini in data selection is as follows:

```
Assume you are an expert in evaluating the accuracy of answers.You will
be provided with a question and two answers:one is the ground truth, and
the other is a model-generated response.You need to score the model's
response based on its similarity to the ground truth, using a scale from 0
to 10.The specific requirements are as follows:

The closer the model's response is to the ground truth, the higher the
score.

1.If there are obvious errors and the model's response is completely
different from the ground truth, score 0-3.

2.If there are errors and the model's response is far from the ground
truth, score 4-6.

3.If there are some errors, and they have some negative impact on the
overall response, score 6-8.

4.If the model's answer is very close to the ground truth, score 9.

5.If the model's response is identical to the ground truth, or even richer
in content and better expressed, score 10.

Please return the score directly in the following format without any
extra information, for example:"Score":"2".
```

## A.2  EXPERIMENTS DETAILS

All single-image experimental results presented in Tab 2 are obtained using the VLMEvalKit (Duan et al., 2024). For the five multi-image benchmarks, MMMU (Yue et al., 2024) is also tested using

| # | | MMMU | BLINK | Mantis | NLVR2 | MVBench | Average |
|---|---|---|---|---|---|---|---|
| | | 35.1 | 37.1 | 41.9 | 52.1 | 36.0 | 40.4 |
| 1 | $\gamma$=0.1 | 35.9 | 41.3 | 46.1 | 53.2 | 39.9 | **43.3** |
| 2 | $\gamma$=0.2 | 37.1 | 39.2 | 42.4 | 51.8 | 39.4 | 42.0 |
| 3 | $\gamma$=0.3 | 35.8 | 39.8 | 42.9 | 52.0 | 39.7 | 42.0 |
| 4 | epoch=1 | 35.9 | 41.3 | 46.1 | 53.2 | 39.9 | 43.3 |
| 5 | epoch=2 | 37.0 | 38.5 | 45.2 | 52.0 | 39.6 | 42.5 |
| 6 | epoch=3 | 36.3 | 42.9 | 44.2 | 54.2 | 39.5 | **43.4** |

Table 4: **Ablation Studies.** The top row refers to the LLaVA-v1.5 baseline. We conduct experiments about the impact of hyperparameter $\gamma$, and training epochs.

| # | | MMMU | BLINK | Mantis | NLVR2 | MVBench | Average |
|---|---|---|---|---|---|---|---|
| | | 35.1 | 37.1 | 41.9 | 52.1 | 36.0 | 40.4 |
| 1 | GPT-Selection | 36.3 | 41.7 | 42.9 | 53.0 | 39.5 | 42.7 |
| 2 | MIA-DPO | 36.3 | 42.9 | 44.2 | 54.2 | 39.5 | 43.4 |
| 3 | $\Delta$ | **0.0** | **+1.2** | **+1.3** | **+1.2** | **0.0** | **+0.7** |

Table 5: **Ablation Studies.** The top row refers to the LLaVA-v1.5 baseline. We conducted an ablation study using GPT-4o-mini for data selection.

VLMEvalKit, while the remaining four multi-image benchmarks, which are not yet fully supported by VLMEvalKit, are tested using the official evaluation code.

During the testing of Mantis, BLINK, and NLVR2, to avoid the model providing irrelevant answers, we add a prompt suffix at the end of the question to guide the model to directly return the multiple-choice option. This makes it easier to extract the answer from the model's response. The prompts we used are listed below: *"Return the choice directly."* or *"Answer:("*

Additionally, when testing multi-image benchmarks, we input the images into the model in sequence. Since the input consists of an image sequence rather than merged images, this significantly increases the length of the image tokens, posing a greater challenge to the model. For Mantis-Eval and MMMU, as they already have well-developed official evaluation codes, we used the official ones for testing.

# B  MODEL AND DATA SOURCES

## B.1  MODEL SOURCES

For the experimental section, we present the testing results of multiple LVLMs on several multi-image and single-image benchmarks. The models involved in the experiments are listed in Tab. 6 of the paper.

## B.2  BENCHMARK SOURCES

The benchmarks involved in the experiments are diverse and include 5 multi-image benchmarks and 7 single-image benchmarks. These benchmarks cover various domains, allowing for a comprehensive assessment of the models' actual capabilities. We list all the benchmarks and their detailed information in Tab. 7, along with a further introduction to some of the benchmarks:

**MMMU**  MMMU (Yue et al., 2024) is a benchmark for assessing multimodal models on college-level tasks that require advanced reasoning and domain-specific knowledge. It features 11,500 questions across six disciplines and includes diverse image types. Initial evaluations show that even advanced model GPT-4V struggles, achieving only 56% accuracy, indicating substantial room for improvement. In addition, MMMU includes both single-image and multi-image test questions.

**BLINK**  BLINK (Fu et al., 2024) is a benchmark for multimodal language models (LLMs) that tests core visual perception tasks solvable by humans "within a blink," like depth estimation and visual correspondence. It reformats 14 classic computer vision tasks into 3,807 multiple-choice questions with images. While humans achieve 95.70% accuracy, top models like GPT-4V and Gemini perform significantly worse, highlighting a gap in visual perception abilities among current LLMs.

**NLVR2**  NLVR2 (Suhr et al., 2018) is a dataset designed for joint reasoning involving natural language and images, focusing on semantic diversity and visual reasoning challenges. It contains

Table 6: **Model Sources.** We have compiled a list of all the models involved in the experiments along with their sources.

| Models | Parameter | Release Time | Source |
|---|---|---|---|
| GPT-4V (Achiam et al., 2023) | - | 2023-09 | Source Link: OpenAI |
| Kosmos2 (Peng et al., 2023) | 1.6B | 2023-06 | Source Link: Kosmos2 |
| VideoLLaVA (Lin et al., 2023) | 7B | 2023-11 | Source Link: Video-LLaVa |
| Fuyu (Bavishi et al., 2023) | 8B | 2023-10 | Source Link: Fuyu-8B |
| VILA (Lin et al., 2024) | 8B | 2023-12 | Source Link: VILA |
| Otter-Image (Li et al., 2023a) | 9B | 2023-05 | Source Link: Otter |
| Idefics1 (Laurençon et al., 2024a) | 9B | 2023-08 | Source Link: Idefics1 |
| BLIP-2 (Li et al., 2023b) | 13B | 2023-01 | Source Link: BLIP-2 |
| OpenFlamingo (Awadalla et al., 2023b) | 9B | 2023-08 | Source Link: OpenFlamingo |
| InstructBLIP (Dai et al., 2023) | 13B | 2023-05 | Source Link: InstructBLIP |
| Qwen-VL-Chat (Bai et al., 2023) | 7B | 2023-8 | Source Link: Qwen-VL-Chat |
| Emu2-Chat (Sun et al., 2024) | 37B | 2023-12 | Source Link: Emu2-Chat |
| CogVLM (Wang et al., 2023) | 17B | 2023-10 | Source Link: CogVLM |
| Idefics2 (Laurençon et al., 2024b) | 8B | 2024-04 | Source Link: Idefics2 |
| LLaVA-v1.6 (Li et al., 2024b) | 7B | 2024-01 | Source Link: LLaVa-Next11 |
| LLaVA-v1.5 (Liu et al., 2024a) | 7B | 2023-10 | Source Link: LLaVa-v1.5 |
| InternLM-XC2.5 (Zhang et al., 2024) | 7B | 2024-07 | Source Link: InternLM-XC2d5 |

Table 7: **Benchmark Sources.** We have included information and links for all the multi-image and single-image benchmarks tested in the paper in the table.

| Setting | Models | Evaluation Metric | Number | Source |
|---|---|---|---|---|
| **Multi-Images Benchmark** | MMMU (Yue et al., 2024) | Multiple Choice | 1,050 | MMMU |
| | BLINK (Fu et al., 2024) | Multiple Choice | 3,807 | BLINK |
| | NLVR2 (Suhr et al., 2018) | Multiple Choice | 6,967 | NLVR2 |
| | Mantis-Eval (Jiang et al., 2024) | Multiple Choice | 217 | Mantis-Eval |
| | MVBench (Li et al., 2024c) | Multiple Choice | 4,000 | MVBench |
| **Single-Image Benchmark** | MMStar (Chen et al., 2024a) | Multiple Choice | 1,500 | MMStar |
| | Sci-QA (Lu et al., 2022) | Multiple Choice | 4,241 | ScienceQA |
| | MMVet (Yu et al., 2023) | Subjective Questions | 218 | MM-Vet |
| | POPE (Li et al., 2023c) | Yes/No | 9,000 | POPE |
| | MMB (Liu et al., 2023) | Multiple Choice | 1,164 | MMBench |
| | Math (Lu et al., 2023) | Multiple Choice | 6,141 | MathVista |
| | AI2D (Kembhavi et al., 2016) | Multiple Choice | 3,090 | AI2D |

107,292 examples of English sentences paired with web photographs, where the task is to determine the truth of a caption regarding a pair of images.

**Mantis-Eval**   Mantis-Eval (Jiang et al., 2024) comprises 217 reasoning examples involving multiple images, addressing various topics like size perception and weight comparisons. Curated by annotators, the dataset features images sourced from Google Search, accompanied by questions that necessitate a thorough understanding of the image content. It includes both multiple-choice and short-answer formats.

**MVBench**   MVBench (Li et al., 2024c) is a dataset that converts static tasks into dynamic video tasks, requiring diverse temporal abilities, from perception to cognition. It automates the generation of multiple-choice questions from public video annotations, ensuring efficient creation and fair evaluation using ground-truth data. The dataset features 20 examples of temporal tasks.

**MMStar**   MMStar (Chen et al., 2024a) is a high-quality benchmark for evaluating multi-modal performance, addressing issues of unnecessary visual content and data leakage in training. It comprises 1,500 carefully selected samples from an initial pool of 22,401, focusing on six core capabilities with 18 detailed dimensions. Each capability features 250 balanced samples, ensuring a comprehensive assessment of multi-modal models.

Table 8: **DPO Data Statistic.** We listed in the table the data volume used for DPO with LLaVa-v1.5 and InternLM-XC2d5, along with the proportion of each type of data.

| Models | Total | Sequence | Grid Collage | Pic-in-Pic |
|---|---|---|---|---|
| LLaVa-v1.5 (Liu et al., 2024a) | 28.9k | 15.1k | 9.3k | 4.5k |
| InternLM-XC2d5 (Zhang et al., 2024) | 23.1k | 11.7k | 7.8k | 3.6k |

**ScienceQA**   ScienceQA (Lu et al., 2022) is a newly collected dataset designed for science question answering, comprising 21,208 multiple-choice questions. Each question includes a multimodal context, the correct option, general background knowledge, and a specific explanation, enabling models to demonstrate multi-step reasoning and interpretability. This dataset addresses the limitations of existing resources by providing detailed explanations alongside the answers.

**MMVet**   MM-Vet (Yu et al., 2023) is designed to evaluate the capabilities of versatile models that integrate various core visual language (VL) functions for solving complex tasks. It defines six key VL abilities—recognition, OCR, knowledge, language generation, spatial reasoning, and mathematical computation—and examines 16 interesting combinations of these functions. The evaluation employs a large language model (LLM)-based open-output assessor, which produces a unified scoring metric across different question types and answer styles.

**POPE**   POPE (Li et al., 2023c) is a dataset designed to evaluate large vision language models (LVLMs) by first extracting ground-truth objects from input images using human annotations or automatic segmentation tools. It then conducts negative sampling for non-existent objects under various settings, including Random, Popular, and Adversarial. Finally, these objects are organized into question templates to assess the models' performance.

**MMBench**   MMBench (Liu et al., 2023) is a benchmark designed to evaluate vision-language (VL) models by addressing the limitations of traditional evaluation methods. It features approximately 3,000 multiple-choice questions across 20 fine-grained ability dimensions, enabling a more comprehensive assessment of model performance. By utilizing ChatGPT to match model predictions with question choices, MMBench ensures robust evaluations that are less biased and more reproducible.

**MathVista**   MathVista (Lu et al., 2023) is a benchmark designed to assess the mathematical reasoning capabilities of large language and multimodal models in visual contexts. It includes 6,141 examples sourced from 28 existing multimodal datasets and three newly created datasets: IQTest, FunctionQA, and PaperQA. The tasks in MathVista require fine-grained visual understanding and compositional reasoning, posing significant challenges for state-of-the-art models.

**AI2D**   AI2D (Kembhavi et al., 2016) is a dataset focused on diagram interpretation and reasoning, addressing the challenge of understanding complex diagrams and their relationships. It includes over 5,000 diagrams and 15,000 annotated questions and answers, providing extensive annotations of diagram constituents and their semantics.

### B.3   MIA-DPO DATA STATISTIC

In constructing our MIA-DPO dataset with three types of multi-image data (Sequence Data, Grid Collage Data, and Pic-in-Pic Data), we used the LLaVa665k (Liu et al., 2024b) dataset as the foundational single-image data. The LLaVa665k dataset contains 665k training samples, with a negligible amount of pure text data, sourced from TextVQA (Singh et al., 2019), COCO (Lin et al., 2014), GQA (Hudson & Manning, 2019), OCRVQA (Mishra et al., 2019), and others. By directly using the existing LLaVa665k dataset, we avoided the high costs of building a multi-image dataset from scratch for MIA-DPO.

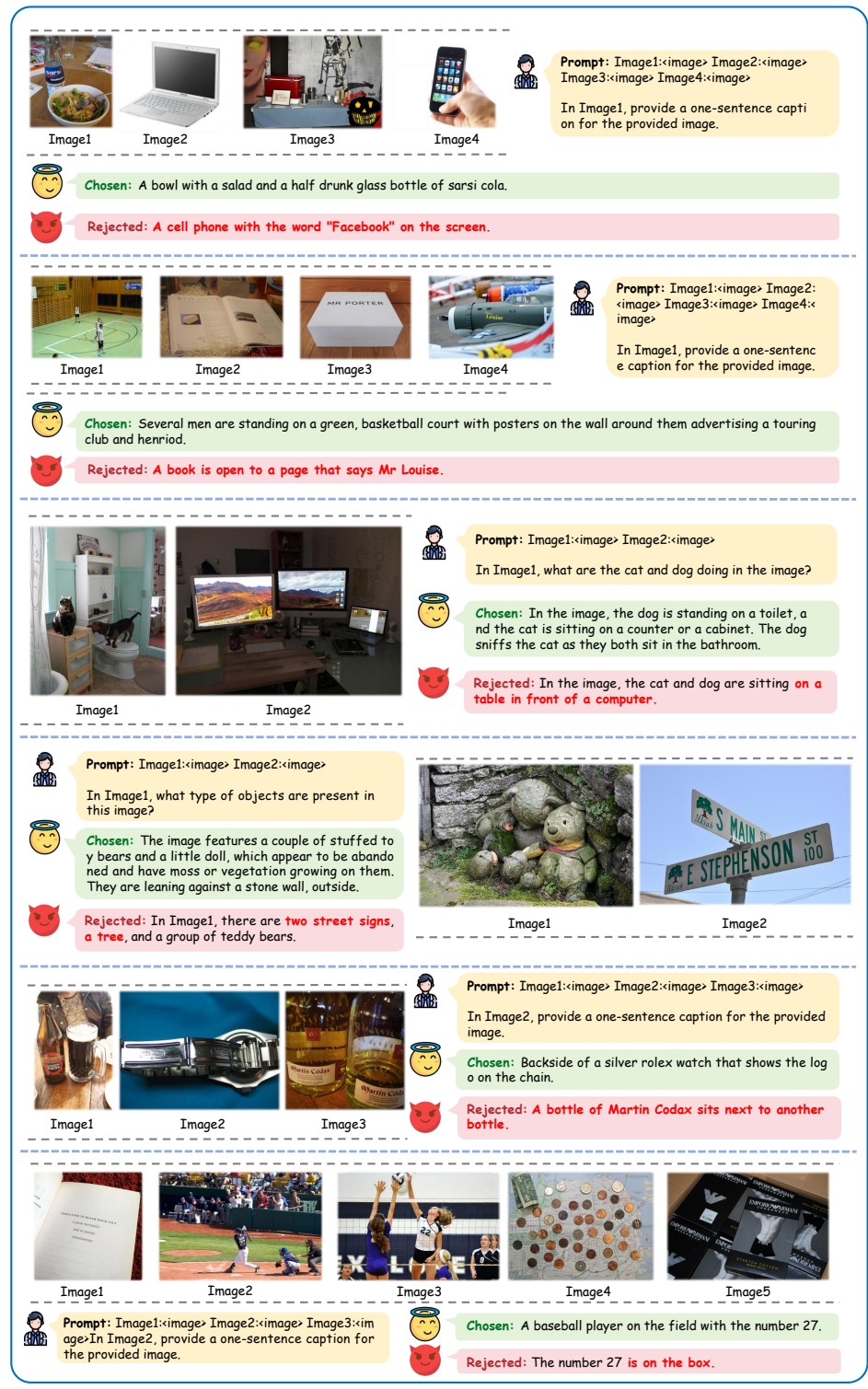

Figure 7: **Sequence Image Data Cases.** The image displays several examples of Sequence data.

Considering that DPO requires only a small amount of data, we randomly sampled a portion of the LLaVa665k dataset to construct the three types of data and employed attention selection in MIA-DPO for filtering. The final data volume used for DPO is summarized in Tab. 8.

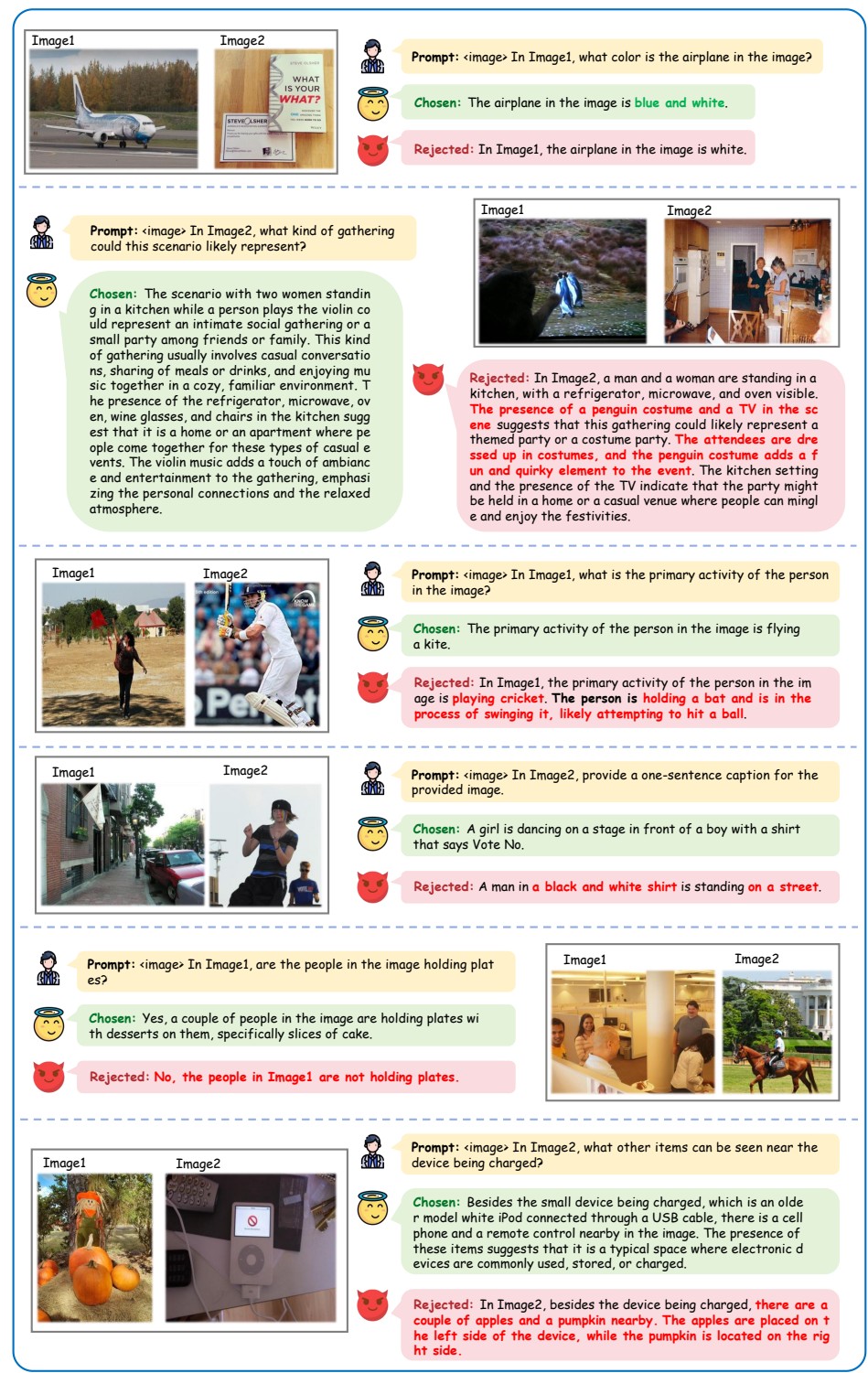

Figure 8: **Grid Collage Data Cases with Two Images.** We present some examples of Grid Collage Data, which consists of images created by stitching together 2 to 9 pictures. Here, we showcase examples of images that combine two pictures.

From Tab. 8, we can see that InternLM-XC2.5 has inherent multi-image data in its pre-training set, which enhances its multi-image capabilities compared to LLaVa-v1.5. As a result, InternLM-XC2.5

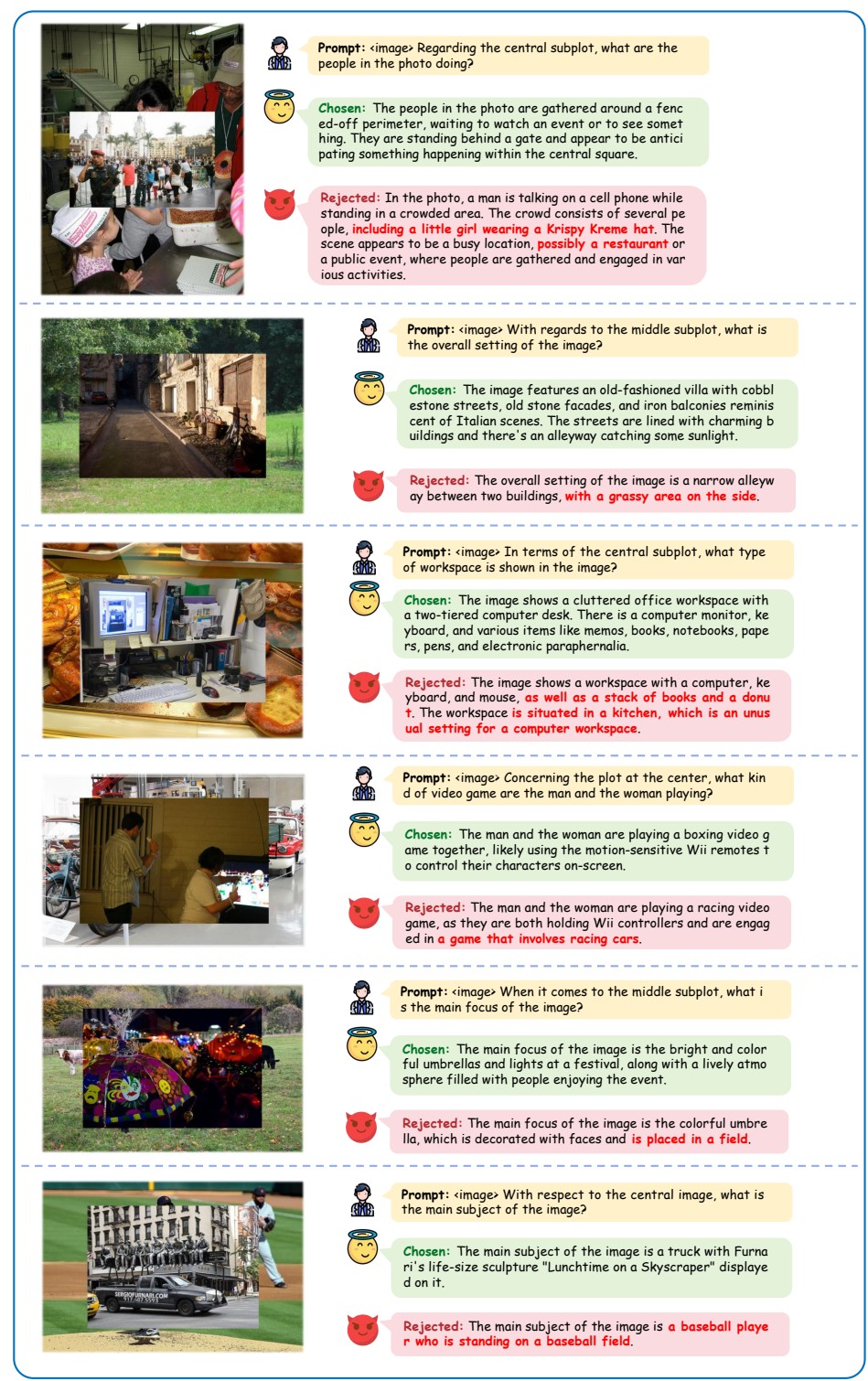

Figure 9: **Pic-in-Pic Image Data Cases.** The image displays several examples of Pic-in-Pic data.

exhibits better attention-aware abilities, leading to a smaller amount of DPO data selected through attention-aware filtering compared to LLaVa-v1.5.

# C  MIA-DPO DATA CASES

## C.1  SEQUENCE IMAGE DATA

Sequence Image Data is the first type of MIA-DPO data we constructed, where multiple images are combined into a sequence, and questions are posed about a randomly selected image within that sequence. The number of images included in Sequence Image Data ranges from 2 to 5. This approach increases the difficulty of answering questions for LVLMs by adding interference from other images beyond the one indicated in the instructions. Additionally, inputting multiple images in sequence significantly increases the length of image tokens, posing a greater challenge for LVLMs. At the same time, the Sequence Image Data type is primarily designed to address the Sequence Confusion type of multi-image hallucination, while also mitigating the Element Interference type of hallucination to some extent. We provide several examples of Sequence Image Data in Fig. 7.

## C.2  GRID COLLAGE IMAGE DATA

Grid Collage Image Data is the second type of MIA-DPO data we constructed, where multiple images are stitched together, and each image is assigned a label such as 'Image1' to indicate which image the instructions refer to for the LVLMs. The number of images in the Grid Collage Data ranges from 2 to 9, forming a large image composed of 1 to 3 rows or columns of smaller images. By combining multiple images, Grid Collage Image Data mixes a vast array of visual elements and details, posing high demands on LVLMs. The instructions for Grid Collage Data involve questioning specific visual elements within the image, with other visual elements serving as interference factors. This data type primarily targets the Element Interference type of hallucination, while the numbered labels for each sub-image also assist the model in addressing the Sequence Confusion type of hallucination. We provide several examples of Grid Collage Image Data in Fig. 8.

## C.3  PIC-IN-PIC IMAGE DATA

Pic-in-Pic Image Data is the third type of MIA-DPO data we constructed. We randomly select two images, resizing one to about half the size of the other, and then paste the smaller image in the center of the larger one. The instructions for Pic-in-Pic Image Data involve questioning the central image, while the background image adds numerous visual elements and details that serve as interference. LVLMs need to carefully distinguish the relationships between these images and integrate the correct visual information to generate answers. Pic-in-Pic Image Data is primarily designed to address the Element Interference type of hallucination. We provide several examples of Pic-in-Pic Image Data in Fig. 9.

# D  MORE OBSERVATION

## D.1  ATTENTION OBSERVATION

In the MIA-DPO architecture, a key step is the selection of chosen and rejected data based on attention. The core idea is to filter data according to the attention-aware capability of LVLMs. For Sequence Data, we assess the ratio of attention between the instructed image and all images. For Grid Collage Data, we evaluate the attention ratio between sub-images and the larger image. For Pic-in-Pic Data, we analyze the attention ratio between the central area of the image and the entire image.

To ensure the smooth execution of attention-based filtering, we visualized the image attention distribution at each layer of the LVLMs, as shown in Fig. 10. The attention distribution of images varies dynamically across different layers. In the early layers of the LVLMs, there are no distinct features in the attention distribution of images, and the same is true for the later layers. However, in the middle layers, there are significant differences in attention distribution among different images. At this point, we can observe where the LVLMs' attention is focused and filter out rejected data where the LVLMs did not attend to the correct areas.

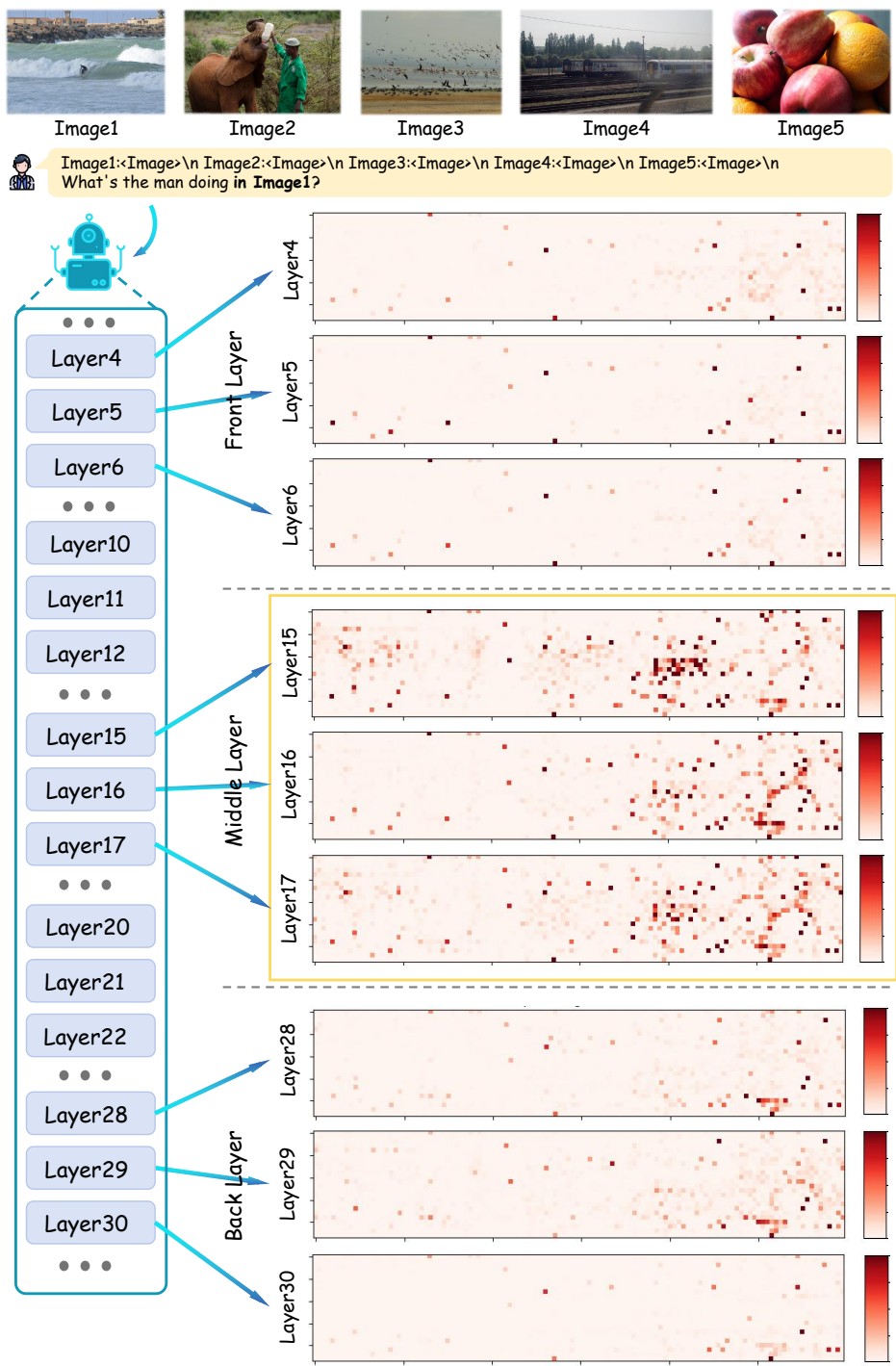

Figure 10: **Attention Observation.** We studied the multi-layer attention of LVLMs and found that the attention of images is most pronounced in the middle layer.

## D.2 HALLUCINATIONS OBSERVATION

In the context of multi-image reasoning, the types of hallucinations that LVLMs may produce are more diverse and varied. Therefore, during the data construction process, we need to specifically analyze these hallucination types. In addition to hallucinations that may occur in single-image tasks, such as existence, attributes, and relation Hallucination, we believe that two unique types of hallucinations may exist in multi-image tasks: Sequence Confusion and Element Interference. These two types of hallucinations are primarily caused by an excessive number of input images that the

Table 9: **Comparison of only use $\mathcal{L}_{\text{NLL}}$ on multi-image benchmarks.**

| Models | Parameter | MMMU | BLINK | Mantis | NLVR2 | MVBench | Average |
|---|---|---|---|---|---|---|---|
| LLaVA-v1.5 | 7B | 35.1 | 37.1 | 41.9 | 52.1 | 36.0 | 40.4 |
| $+ \mathcal{L}_{\text{NLL}}$ | 7B | 36.1 | 39.1 | 43.5 | 51.8 | 39.9 | 42.1 (**+1.7**) |
| $+ \mathcal{L}_{\text{DPO}} + \mathcal{L}_{\text{NLL}}$ | 7B | 36.3 | 42.9 | 44.2 | 54.2 | 39.5 | **43.4** (**+3.0**) |

Table 10: **Comparison of only use $\mathcal{L}_{\text{NLL}}$ on single-image benchmarks.**

| Models | Parameter | MMStar | SQA | MMVet | POPE | MMB | Math | AI2D | Average |
|---|---|---|---|---|---|---|---|---|---|
| LLaVA-v1.5 | 7B | 32.9 | 66.6 | 30.5 | 85.9 | 64.3 | 25.4 | 55.5 | 51.6 |
| $+ \mathcal{L}_{\text{NLL}}$ | 7B | 33.7 | 67.6 | 28.2 | 76.9 | 61.8 | 24.9 | 56.3 | 49.9 (**-1.7**) |
| $+ \mathcal{L}_{\text{DPO}} + \mathcal{L}_{\text{NLL}}$ | 7B | 32.9 | 67.6 | 32.1 | 87.2 | 63.1 | 24.4 | 54.7 | **51.7** (**+0.1**) |

LVLMs cannot follow in sequence, as well as the overwhelming number of image tokens and visual elements.

In the process of constructing DPO data, we take hallucination types as our starting point and thoroughly consider solutions for these two types of hallucinations. More hallucination cases are already presented in Fig. 7, Fig. 8, Fig. 9.

## E    FURTHER EXPLORATION

### E.1    COMPARISON WITH $\mathcal{L}_{\text{NLL}}$ ONLY

The $\mathcal{L}_{\text{NLL}}$ term in Eq. (5) plays a role of supervised fine-tuning (SFT) with only chosen answers. We compare the baseline of fine-tuning with only the $\mathcal{L}_{\text{NLL}}$, and results are presented in Tab. 9 and 10 for multi-image and single-image benchmarks, respectively.

On multi-image benchmarks (Tab. 9), fine-tuning merely with the $\mathcal{L}_{\text{NLL}}$ (second row) yields a performance improvement over the LLaVA-v1.5 baseline. However, MIA-DPO (third row) consistently outperforms the $\mathcal{L}_{\text{NLL}}$-only baseline, demonstrating the significant contribution of negative samples to model improvement.

On single-image benchmarks (Tab. 10), the $\mathcal{L}_{\text{NLL}}$-only baseline (second row) leads to a performance degradation compared to LLaVA-v1.5, highlighting the potential risks of incorporating multi-image data during SFT may adversely affect performance on single-image tasks. By contrast, MIA-DPO (third row) maintains performance parity with LLaVA-v1.5, thanks to the KL-divergence loss constraint in Eq. (3). This further demonstrates the advantages of MIA-DPO over the $\mathcal{L}_{\text{NLL}}$-only baseline.

### E.2    COMPARISON WITH SFT ON MULTI-IMAGE DATA WITHOUT DPO

We conducted experiments comparing MIA-DPO with existing multi-image SFT methods including Mantis (Jiang et al., 2024) or MMDU (Liu et al., 2024d). The results, presented in Tab. 11 and 12, demonstrate that MIA-DPO effectively improves multi-image performance without compromising performance on single-image benchmarks. In contrast, direct SFT on multi-image data can lead to a slight degradation in single-image performance. This highlights the advantage of MIA-DPO in maintaining a balance between both tasks.

### E.3    ABLATION STUDIES ON THRESHOLD AND DATA RATIOS

**Threshold** We conduct multiple ablation experiments with different threshold ranges, see Tab. 13 and Tab. 14. We observe that our MIA-DPO is relatively robust to different threshold ranges, and our default choices (0.7/0.6/0.5/0.5) perform slightly better than other choices.

Additionally, we present the baseline that uses uniform thresholds, and results are shown in the fourth row of Tab. 13 and Tab. 14. The experimental results demonstrate that using a uniform

Table 11: **Comparison of SFT on multi-image data** with MIA-DPO on multi-image benchmarks.

| Models | Parameter | MMMU | BLINK | Mantis | NLVR2 | MVBench | Average |
|---|---|---|---|---|---|---|---|
| LLaVA-v1.5 | 7B | 35.1 | 37.1 | 41.9 | 52.1 | 36.0 | 40.4 |
| + SFT on MMDU | 7B | 36.1 | 39.2 | 42.9 | 53.7 | 39.1 | 42.2 |
| + SFT on Mantis | 7B | 35.9 | 39.5 | **45.6** | 52.8 | **40.9** | 42.9 |
| + MIA-DPO (Ours) | 7B | **36.3** | **42.9** | 44.2 | **54.2** | 39.5 | **43.4** |

Table 12: **Comparison of SFT on multi-image data** with MIA-DPO on single-image benchmarks.

| Models | Parameter | MMStar | SQA | MMVet | POPE | MMB | Math | AI2D | Average |
|---|---|---|---|---|---|---|---|---|---|
| LLaVA-v1.5 | 7B | 32.9 | 66.6 | 30.5 | 85.9 | 64.3 | 25.4 | 55.5 | 51.6 |
| + SFT on MMDU | 7B | 32.5 | 65.7 | 30.4 | 85.9 | 62.3 | 25.6 | 54.7 | 51.0 |
| + SFT on Mantis | 7B | 32.9 | 65.9 | 29.8 | 84 | 63.5 | 25.9 | 54.4 | 50.9 |
| + MIA-DPO (ours) | 7B | 32.9 | 67.6 | 32.1 | 87.2 | 63.1 | 24.4 | 54.7 | 51.7 |

threshold can negatively impact performance. This observation is due to the attention ratio distributions varying significantly depending on the number of images (see Fig. 5), which indicates that a one-size-fits-all threshold is not optimal. By adjusting the threshold based on the number of images, we can improve the model's ability to generate accurate and coherent responses.

**Data Ratios** We present ablation studies on different data proportions in Tab. 13 and Tab. 14. We adjust the proportions of the three data types and evaluate the results. The results indicate that the model's performance on both multi-image and single-image tasks remains stable, with only minor fluctuations. This demonstrates the robustness of our model across different data proportion settings. In conclusion, none of the data types showed a particularly significant advantage over others. When used in combination, the three data types achieve better performance than any single type used individually.

### E.4 ABLATION STUDIES ON MODEL SIZE

To explore the effectiveness of larger models, we apply MIA-DPO with the LLaVa-v1.5-13B model. We observe that the attention distribution patterns of the 13B model are largely consistent with those of the 7B model. The results of MIA-DPO + LLaVa-v1.5-13B on multi-image and single-image benchmarks are presented in Tab. E.7 and Tab. 17. These results further validate our conclusions about the effectiveness of MIA-DPO, demonstrating that it can consistently improve the performance of larger-size models.

### E.5 VQA TEST SET AND LARGER IMAGE NUMBERS

We construct a VQA test set of 500 questions using images and questions from LLaVA-665k but are mutually exclusive with the MIA-DPO training data. This test set includes questions with 2 to 5 images per question, allowing us to directly assess improvements in sequence confusion and element interference. By evaluating the pre- and post-DPO versions of LLaVA and IXC2.5 on this test set, we observed accuracy improvements of 5.8% and 1.9%, respectively (see Tab. 18). These results further validate the effectiveness of MIA-DPO in enhancing multi-image understanding. We plan to release this VQA test set in the final version of our paper to facilitate future research in this area.

Additionally, We constructed a new VQA test set (the construction steps keep the same) consisting of 50 questions each for 4, 6, 8, and 10 images. We report the performance of IXC 2.5 + MIA-DPO in Tab. 19. Our MIA-DPO consists of improving the multi-image understanding abilities as the number of images increases. However, the performance of LVLMs, such as IXC2.5, on extremely large numbers of images will also be limited by factors like context window size. As a result, the performance gains from MIA-DPO will gradually diminish with an increasing number of images. These findings show potential for future research on long-context abilities, such as ROPE extrapolation on LVLMs.

Table 13: **Ablation studies of attention threshold and data ratio** on multi-image benchmarks.

| # | Models | Setting | MMMU | BLINK | Mantis | NLVR2 | MVBench | Average |
|---|---|---|---|---|---|---|---|---|
| | LLaVA-v1.5 | - | 35.1 | 37.1 | 41.9 | 52.1 | 36.0 | 40.4 |
| 1 | 0.8/0.7/0.6/0.6 | threshold | 37.5 | 40.5 | 42.4 | 54.3 | 40.3 | 43.0 |
| 2 | 0.7/0.6/0.5/0.5 | threshold | 36.3 | 42.9 | 44.2 | 54.2 | 39.5 | **43.4** |
| 3 | 0.6/0.5/0.4/0.4 | threshold | 36.2 | 41.1 | 44.7 | 54.1 | 39.9 | 43.2 |
| 4 | 0.6/0.6/0.6/0.6 | threshold | 36.2 | 40.6 | 41.9 | 53.7 | 39.2 | 42.3 |
| 5 | 2:3:1 | data ratio | 36.1 | 41.5 | 42.9 | 55.0 | 39.4 | 43.0 |
| 6 | 2:1:3 | data ratio | 37.1 | 39.8 | 46.0 | 52.4 | 40.2 | 43.2 |
| 7 | 3:2:1 | data ratio | 36.3 | 42.9 | 44.2 | 54.2 | 39.5 | **43.4** |

Table 14: **Ablation studies on threshold and data ratio** on single-image benchmark.

| # | Models | Setting | MMStar | SQA | MMVet | POPE | MMB | Math | AI2D | Average |
|---|---|---|---|---|---|---|---|---|---|---|
| | LLaVA-v1.5 | - | 32.9 | 66.6 | 30.5 | 85.9 | 64.3 | 25.4 | 55.5 | 51.6 |
| 1 | 0.8/0.7/0.6/0.6 | threshold | 32.2 | 67.7 | 31.1 | 86.8 | 63.8 | 25.0 | 55.1 | 51.7 |
| 2 | 0.7/0.6/0.5/0.5 | threshold | 32.9 | 67.6 | 32.1 | 87.2 | 63.1 | 24.4 | 54.7 | **51.7** |
| 3 | 0.6/0.5/0.4/0.4 | threshold | 33.1 | 67.3 | 32.2 | 87.1 | 63.2 | 24.6 | 54.7 | 51.7 |
| 4 | 0.6/0.6/0.6/0.6 | threshold | 33.0 | 67.1 | 32.3 | 86.4 | 62.5 | 25.5 | 54.7 | 51.6 |
| 5 | 2:3:1 | data ratio | 32.2 | 67.0 | 32.6 | 87.1 | 63.2 | 24.5 | 54.6 | 51.6 |
| 6 | 2:1:3 | data ratio | 32.1 | 67.5 | 32.0 | 87.4 | 63.3 | 24.7 | 54.7 | 51.6 |
| 7 | 3:2:1 | data ratio | 32.9 | 67.6 | 32.1 | 87.2 | 63.1 | 24.4 | 54.7 | **51.7** |

## E.6 MORE BASELINES OF INTERNLM-XCOMPOSER2.5

Since the authors of POVID and HA-DPO have open-sourced their trained models based on LLaVa, we directly tested their released models and report in Tab. 1 and Tab. 2. Additionally, we conducted additional experiments by applying DPO to InternLM-XC2.5 using the datasets provided by POVID and HA-DPO. As shown in Tab. 1 and Tab. 2, the performance improvements for applying POVID and HA-DPO on multi-image benchmarks were limited. This suggests that while these previous papers are designed for single-image scenarios, their direct application to multi-image settings may not yield significant gains. We believe that the limitations observed in these experiments highlight the unique challenges posed by multi-image hallucinations and the need for specialized techniques like our proposed MIA-DPO approach.

## E.7 ANALYSIS OF FAILURE CASES

We provide some failure cases in Fig. 11. The cases presented are from the multi-image QA of the MMMU benchmark. We observe that although MIA-DPO has improved the model's multi-image understanding and reasoning capabilities, the model may still make errors when encountering questions from out-of-domain knowledge (e.g., fine-grained plant classification, medical image processing), which is not present in our training data. We believe extending the training data to more diverse domains will alleviate these failure cases.

Table 15: **Ablation Study on Different Components of Post-Selection.** We conducted ablation experiments on different components of post-selection to explore their respective roles and contributions.

| # | Models | MMMU | BLINK | Mantis | NLVR2 | MVBench | Average |
|---|--------|------|-------|--------|-------|---------|---------|
|   | LLaVA-v1.5 | 35.1 | 37.1 | 41.9 | 52.1 | 36.0 | 40.4 |
| 1 | w/o post sel. | 35.3 | 38.7 | 44.2 | 53.7 | 39.4 | 42.3 |
| 2 | w ppl | 35.6 | 40.4 | 44.2 | 53.8 | 39.5 | 42.7 |
| 3 | w length ratio | 35.7 | 40.6 | 44.3 | 53.7 | 39.4 | 42.7 |
| 4 | w edit distance | 35.6 | 40.0 | 44.2 | 53.9 | 39.4 | 42.6 |
| 5 | MIA-DPO | 36.3 | **42.9** | 44.2 | 54.2 | 39.5 | **43.4** |

Table 16: **Study on Model Size for Multi-Image Benchmarks.** We conduct MIA-DPO on larger model LLaVa-v1.5-13B (Liu et al., 2024b).

| Models | Parameter | MMMU | BLINK | Mantis | NLVR2 | MVBench | Average |
|--------|-----------|------|-------|--------|-------|---------|---------|
| LLaVA-v1.5 | 7B | 35.1 | 37.1 | 41.9 | 52.1 | 36.0 | 40.4 |
| + MIA-DPO (Ours) | 7B | **36.3** | **42.9** | **44.2** | **54.2** | **39.5** | **43.4** |
| Δ | - | +1.2 | +5.8 | +2.3 | +2.1 | +3.5 | +3.0 |
| LLaVA-v1.5 | 13B | 37.0 | 40.9 | **47.0** | 62.5 | 40.6 | 45.6 |
| + MIA-DPO (Ours) | 13B | **38.8** | **42.4** | 46.5 | **64.5** | **42.0** | **46.8** |
| Δ | - | +1.8 | +1.5 | -0.5 | +2.0 | +1.4 | +1.2 |

Table 17: **Study on Model Size for Single-Image Benchmark.** We conduct MIA-DPO on larger model LLaVa-v1.5-13B (Liu et al., 2024b).

| Models | Parameter | MMStar | SQA | MMVet | POPE | MMB | Math | AI2D | Average |
|--------|-----------|--------|-----|-------|------|-----|------|------|---------|
| LLaVA-v1.5 | 7B | 32.9 | 66.6 | 30.5 | 85.9 | 64.3 | 25.4 | 55.5 | 51.6 |
| + MIA-DPO (ours) | 7B | 32.9 | 67.6 | 32.1 | 87.2 | 63.1 | 24.4 | 54.7 | 51.7 |
| LLaVA-v1.5 | 13B | 34.3 | 71.2 | 35.6 | 86.7 | 68.5 | 27.7 | 61.1 | 55.0 |
| + MIA-DPO (ours) | 13B | 33.4 | 69.7 | 36.9 | 85.1 | 68.2 | 25.1 | 59.4 | 54.0 |

| # | Accuracy | Image Numbers | LLaVa-v1.5 | IXC2.5 |
|---|----------|---------------|------------|--------|
| 1 | Original | 2-5 | 75.8 | 88.4 |
| 2 | MIA-DPO | 2-5 | 81.6 | 90.3 |
| 3 | Δ | 2-5 | +5.8 | +1.9 |

Table 18: **Results on Our Own VQA Test Set.** Our VQA set consists of 500 questions, each involving 2 to 5 images.

| # | Models | 4-Images | 6-Images | 8-Images | 10-Images |
|---|--------|----------|----------|----------|-----------|
| 1 | IXC2.5 | 76 | 74 | 68 | 10 |
| 2 | + MIA-DPO | 86 | 82 | 72 | 12 |
| 3 | Δ | +10 | +8 | +4 | +2 |

Table 19: **Results with more images.** MIA-DPO continues to improve multi-image understanding abilities as the number of images increases, up to the limit of the context window.

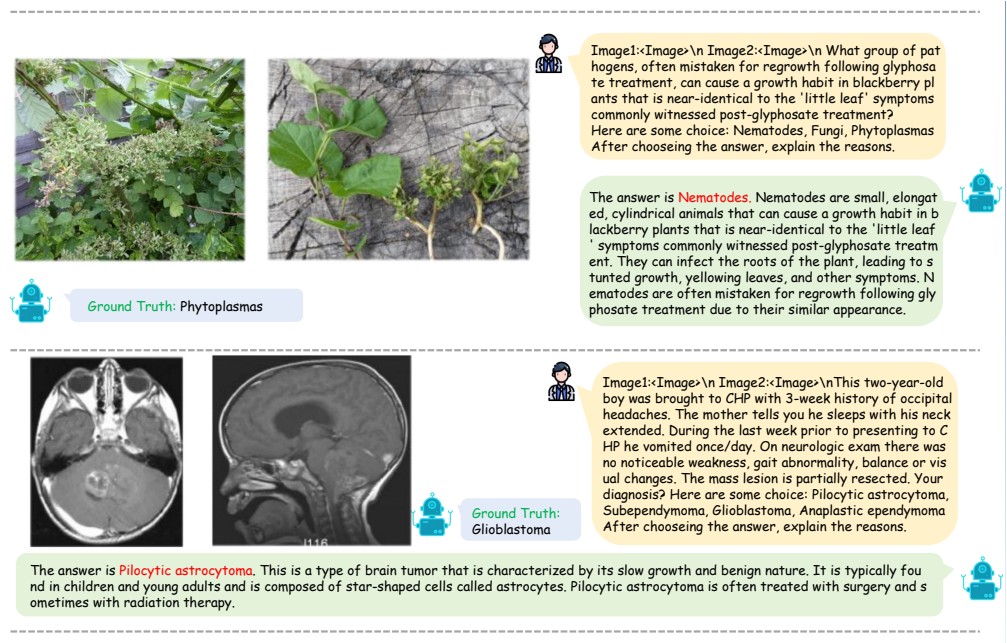

Figure 11: **Analysis of Failure Cases.** Although MIA-DPO has improved the model's multi-image understanding and reasoning capabilities, the model may still make errors when encountering questions from out-of-domain unseen in training data (e.g., fine-grained plant classification, medical image processing).

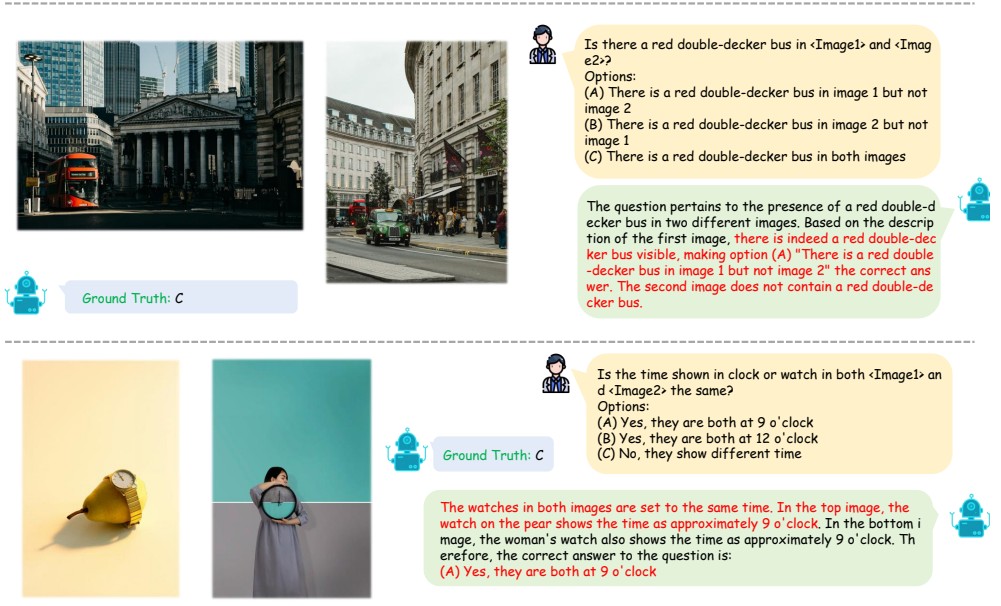

Figure 12: **Analysis of Mantis Benchmark.** Original IXC2.5 provides lengthy descriptions instead of concise answers for multiple-choice questions.

