# OpenReview forum: "MIA-DPO: Multi-Image Augmented Direct Preference Optimization For Large Vision-Language Models"
_ICLR.cc/2025/Conference — ICLR 2025 Poster_

### Official Review · Reviewer_DZHj · 2024-11-03

**Soundness:** 2
**Presentation:** 2
**Contribution:** 2
**Rating:** 6
**Confidence:** 3

**Summary:**

This paper proposes MIA-DPO, a method for improving the Large Vision Language Modle (LVLM) with multiple images. The main idea of MIA-DPO is finding the rejected prompts by using attention from pretrained LVLM and using the DPO training to fine-tune the LVLM for multiple image settings. Through the proposed MIA-DPO, the paper achieves better performance on multiple image settings than the baseline LVLM, without needing 1) The data for the multiple image setting and 2) external sources such as human/ChatGPT for creating feedback in the RLHF/RLAIF methods.

**Strengths:**

- The motivation of this paper is clearly stated, which helps the reader a lot in understanding the idea.

- The proposed **Attention Aware Selection** can alleviate the problem of multiple image prompt datasets requirement and the need for external human/module in RLHF/RLAIF approaches.

- The experiments show the effectiveness of the method against the baselines.

**Weaknesses:**

**Clarification**:
- Can the authors provide a more specific way to compute $A_{target}$ and $A_{sum}$ in L306? In L306-L307, the paper states that "$A_{target}$ be the amount of attention directed toward the target defined in $x$", however, as defined in L150, $x$ consists of $v$ and $t$, and as mentioned in Figure 2, only the attention on the given images $v$ is considered.
- Which are the layers used to extract the attention $A_{target}$ and $A_{sum}$?
- In section 3.3.3, what are the parameters we need to optimize in the proposed method?

**Experiments**:
- In Table 1 and Table 2, why do the authors not include "RLHF", "HA-DPO" and "POVID" in InternLM-XC2.5?
- In Table 1, why LLaVA-RLHF, HA-DPO, and POVID are worse than LLaVA?

**Questions:**

All the concerns and questions are listed in the Weaknesses section.

---

> ### Author Response · Authors · 2024-11-21
> **Authors Rebuttal to Reviewer DZHj**
>
> Dear Reviewer DZHj,
>
> We sincerely thank you for your thorough feedback: the motivation of this paper is clearly stated and the effectiveness of the method. We address your questions below and have incorporated all feedback in the revised version. ***All new material added to the revised manuscript has been highlighted in red text for better visibility.***
>
> > Q1: Can the authors provide a more specific way to compute $A_{target}$ and $A_{sum}$ in L306?
>
> A1: Apologies for not clearly explaining the calculation methods for $A_{sum}$ and  $A_{target}$. As described in Line 150, $x$ represents the input, which includes images $v$ and text $t$, where $v$ may consist of multiple images.
> When $x$ is input into the model, we calculate the attention values of the model's output and the text $t$ tokens relative to the image $v$ tokens, obtaining the attention value for each image. The total attention of the image $v$ pointed to by the instruction (i.e., $t$) is defined as $A_{target}$, while the total attention across all images is defined as $A_{sum}$. In Figure 2, taking the first example as an illustration, we input five images and visualized the concatenated attention map of the five images. The total attention for the first image corresponds to $A_{target}$, and the total attention for all five images corresponds to $A_{sum}$.
>
> > Q2: Which are the layers used to extract the attention $A_{target}$ and $A_{sum}$?
>
> A2: Good question! During our experiments, we observed an interesting phenomenon: the differences in attention distribution across different images are most pronounced in the intermediate layers of the model, while this phenomenon is less evident in the initial and final layers. Therefore, we chose the intermediate layers of the model as the layers for calculating $A_{target}$ and $A_{sum}$.
>
> > Q3: In section 3.3.3, what are the parameters we need to optimize in the proposed method?
>
> A3: The LLaVA-based LVLMs typically have three components: a vision encoder, an MLP projector, and an LLM. During the DPO process, the vision encoder of the LVLM remains frozen, and the remaining parameters (MLP projector, LLM) are optimized.
>
> > Q4: In Table 1 and Table 2, why do the authors not include "RLHF", "HA-DPO" and "POVID" in InternLM-XC2.5?
>
> A4: Thanks for your comment! Since the authors of POVID and HA-DPO have open-sourced their trained models based on LLaVa, we directly tested their released models and report in Table 1 and Table 2. Following your suggestion, we conducted additional experiments by applying DPO to InternLM-XC2.5 using the datasets provided by POVID and HA-DPO. As shown in Table 1, the performance improvements for applying POVID and HA-DPO on multi-image benchmarks were limited. This suggests that while these previous papers are designed for single-image scenarios, their direct application to multi-image settings may not yield significant gains. We believe that the limitations observed in these experiments highlight the unique challenges posed by multi-image hallucinations and the need for specialized techniques like our proposed MIA-DPO approach.
>
> > Q5: In Table 1, why LLaVA-RLHF, HA-DPO, and POVID are worse than LLaVA?
>
> A5: While previous works (LLaVA-RLHF, HA-DPO, and POVID) are effective methods for addressing single-image hallucinations, they do not consider multi-image scenarios. Thus, these methods are often trained on datasets that exhibit certain biases on single-image data and may not align well with the diverse and complex nature of multi-image understanding abilities. So in some cases, the application of these techniques can even degrade the original multi-image capabilities of the underlying LLaVA-v1.5 model.

---

> ### Author Response · Authors · 2024-11-24
>
> Dear Reviewer DZHj,
>
> We deeply appreciate your efforts in reviewing our paper, and we are particularly grateful for the invaluable suggestions that have greatly enhanced its quality.
>
> In response to your suggestions, we provided more details on computing $A_{target}$ and $A_{sum}$ (Q1 and Q2) and introduced which parameters need to be optimized in our method (Q3). Subsequently, we presented additional baselines for IXC2.5, including the performance of POVID and HA-DPO on IXC2.5 (Q4). Finally, we explained why LLaVA-RLHF, HA-DPO, and POVID perform worse than LLaVA.
>
> Could you please read our rebuttal and consider adjusting the score in light of our response? Thank you for your time and effort!
>
> Best,
>
> Authors

---

> ### Author Response · Authors · 2024-11-26
>
> Dear Reviewer DZHj,
>
> Thank you again for your time and valuable feedback on our paper. We have carefully addressed all your comments in our rebuttal and would greatly appreciate it if you could review our response and consider adjusting your score accordingly.
>
> Your support means a lot to us, and we are grateful for your consideration.
>
> Best regards,
>
> Authors

---

> ### Author Response · Authors · 2024-11-27
>
> Dear Reviewer DZHj,
>
> I hope this message finds you well. I’m writing to kindly follow up on the review process for our paper. We deeply value your expertise and would greatly appreciate it if you could provide your feedback and scoring at your earliest convenience to help progress the review.
>
> We humbly hope for your positive consideration and support in evaluating our work. Your recognition of the effort we’ve put into this research would mean a great deal to us.
>
> Thank you so much for your time and thoughtful review.
>
> Best regards,
>
> Authors

---

> ### Author Response · Authors · 2024-11-28
>
> Dear Reviewer DZHj,
>
> I hope this message finds you well. As we are in the final stages, we would greatly appreciate your valuable feedback and scoring at your earliest convenience.
>
> We sincerely understand how busy you must be, and we are deeply grateful for the time and effort you’ve already dedicated to reviewing our paper. Your support and thoughtful feedback are extremely important to us, and we would be thankful for any comments you can provide.
>
> Thank you once again for your time and consideration.
>
> Best regards,
>
> Authors

---

> ### Author Response · Authors · 2024-12-01
>
> Dear Reviewer DZHj.
>
> The deadline is approaching. Now two reviewers post their response to our rebuttal, and our paper received mixed results (one 6 rating and one 5 rating). We have not received any response from you. Can you read our rebuttal and post your comments? Thanks.

---

> > ### Comment · Reviewer_DZHj · 2024-12-02
> >
> > Thanks for your response. The computation of $A_{target}$ and $A_{sum}$ is clear to me. The concern of the baseline is addressed. Therefore, I will increase the point to 6 for this submission.

---

> > > ### Author Response · Authors · 2024-12-02
> > >
> > > Dear Reviewer DZHj,
> > >
> > > Thank you very much for your thoughtful response and for increasing your  score to 6. We are truly grateful for your recognition of the quality and significance of our work. Your feedback and encouragement mean a great deal to us and motivate us to continue improving our research.
> > >
> > > We appreciate your time and effort in thoroughly reviewing our paper, and we are delighted that the revisions have addressed your concerns to your satisfaction.
> > >
> > > Thank you once again for your constructive feedback and for helping to make our paper stronger.
> > >
> > > Sincerely,
> > >
> > > Authors

---

### Official Review · Reviewer_ZLYy · 2024-11-03

**Soundness:** 2
**Presentation:** 2
**Contribution:** 2
**Rating:** 5
**Confidence:** 4

**Summary:**

In this paper, the authors address the performance limitations of existing Large Vision Language Models(LVLMs) in multi-image tasks, identifying the difficulty of constructing multi-image preference data at low cost as a critical constraint. To tackle this issue, the authors propose Multi-Image Augmented Direct Preference Optimization (MIA-DPO), which can efficiently scale single-image training data to multi-image data without relying on human annotations or external APIs. In terms of data construction, MIA-DPO designs three expansion methods—Sequence, Grid Collage, and Pic-in-Pic to extend single-image data into multi-image version. Additionally, to address the common visual hallucinations of sequence confusion and element interference in  multi-image tasks, a hallucination detection mechanism based on attention values is designed for the constructed multi-image data. The authors conducted ablation experiments on LLaVA-v1.5 and InternLM-XC2.5, and the results demonstrate that the proposed MIA-DPO can enhance performance on multi-image tasks such as MVBENCH without compromising single-image metrics like MMMU.

**Strengths:**

1. Research on visual hallucinations in large vision-language models is significant as it can improve the general capabilities of these models.
2. This paper presents an efficient pipeline for constructing and filtering multi-image preference data, which can generate large amounts of multi-image preference data at a low cost, making it highly applicable.
3. The paper analyzes the prevalent sequence confusion and element interference phenomena in large vision-language models for multi-image tasks from the perspective of attention, identifying that the key influential factor of these multi-image hallucinations is incorrect attention distribution.

**Weaknesses:**

1. The authors mention at the beginning of the paper that existing multi-image SFT methods can degrade performance on single-image tasks while improving multi-image performance. However, they do not compare their approach with these methods in the experimental section.
2. Although the MIA-DPO method can easily convert data into SFT format, the authors do not compare the performance changes of the model after SFT using the same source data.
3. When evaluating the model's single-image performance, only MMMU is used, without assessing performance on other widely recognized benchmarks such as MMStar, MMBench, and OCRBench.
4. For evaluating the model's multi-image performance, the authors could use the Sequence, Grid Collage, and Pic-in-Pic methods to construct multi-image VQA test sets from existing single-image VQA data, directly assessing improvements in sequence confusion and element interference.
5. While handling more images is crucial for large vision-language models, the paper does not analyze whether MIA-DPO can still provide stable benefits when the number of simultaneously appearing images increases significantly (e.g., 10+ images).

**Questions:**

1. How does the model's performance change on other single-image or multi-image metrics before and after applying MIA-DPO? (Refer to Weaknesses)
2. Can MIA-DPO still provide stable benefits when the number of images is expanded? (Refer to Weaknesses)
3. Can experiments be conducted on more advanced open-source large vision-language models or those explicitly supporting multi-images, such as the Qwen-VL series and Tarsier?

---

> ### Author Response · Authors · 2024-11-21
> **Authors Rebuttal to Reviewer ZLYy (1/2)**
>
> Dear Reviewer ZLYy,
>
> We sincerely thank you for your thorough feedback:  an efficient pipeline for constructing and filtering multi-image preference data. We address your questions below and have incorporated all feedback in the revised version. ***All new material added to the revised manuscript has been highlighted in red text for better visibility.***
>
> > Q1: The authors mention at the beginning of the paper that existing multi-image SFT methods can degrade performance on single-image tasks while improving multi-image performance. However, they do not compare their approach with these methods in the experimental section.
>
> A1：Thank you for your valuable feedback. To address your concern, we conducted experiments comparing MIA-DPO with existing multi-image SFT methods, including Mantis and MMDU. The results, presented in Tab. 11 and Tab. 12, demonstrate that MIA-DPO effectively improves multi-image performance without compromising performance on single-image benchmarks. In contrast, direct SFT on multi-image data can lead to a slight degradation in single-image performance. This highlights the advantage of MIA-DPO in maintaining a balance between both tasks.
>
> > Q2: Although the MIA-DPO method can easily convert data into SFT format, the authors do not compare the performance changes of the model after SFT using the same source data.
>
> A2:  Valuable feedback! Based on your insightful suggestion, we have added experiment in Tab. 9 and Tab. 10. We convert the chosen DPO data into the SFT format, and fine-tune the model using merely the NLL loss term in Eq. (5). The results show that, compared to directly SFT with the chosen data (second row), MIA-DPO (third row) achieves more significant performance improvements on multi-image benchmarks, which demonstrate the importance of negative samples in the DPO loss.
>
> > Q3: When evaluating the model's single-image performance, only MMMU is used, without assessing performance on other widely recognized benchmarks such as MMStar, MMBench, and OCRBench.
>
> A3: Since the MMMU is a benchmark that includes both single-image and multi-image QA, we have placed it in Tab. 1. We present the testing results for seven single-image benchmarks (e.g., MMBench, MMStar, SQA, POPE, MMB) in Tab. 2. In response to your suggestion, we have added the results of OCRBench to Tab. 2.
>
> > Q4: For evaluating the model's multi-image performance, the authors could use the Sequence, Grid Collage, and Pic-in-Pic methods to construct multi-image VQA test sets from existing single-image VQA data, directly assessing improvements in sequence confusion and element interference.
>
> A4: Good suggestion! Your feedback is highly valuable to us. We construct a VQA test set of 500 questions using images and questions from LLaVA-665k that are mutually exclusive with the MIA-DPO training data. This test set includes questions with 2 to 5 images per question, allowing us to directly assess improvements in sequence confusion and element interference. By evaluating the pre- and post-DPO versions of LLaVA and IXC2.5 on this test set, we observed accuracy improvements of 5.8% and 1.9%, respectively (see Tab. 18). These results further validate the effectiveness of MIA-DPO in enhancing multi-image understanding. We plan to release this VQA test set in the final version of our paper to facilitate future research in this area.

---

> ### Author Response · Authors · 2024-11-21
> **Authors Rebuttal to Reviewer ZLYy (2/2)**
>
> > Q5: While handling more images is crucial for large vision-language models, the paper does not analyze whether MIA-DPO can still provide stable benefits when the number of simultaneously appearing images increases significantly (e.g., 10+ images).
>
> A5: Thank you for your insightful suggestion. We agree that processing extremely large numbers of images is indeed a valuable direction for future work. We constructed a VQA test set (the construction steps are the same as A4) consisting of 50 questions each for 4, 6, 8, and 10 images. We report the performance of IXC 2.5 + MIA-DPO in Tab. 19. Our MIA-DPO consists of improving the multi-image understanding abilities as the number of images increases. However, the performance of LVLMs, such as IXC2.5, on extremely large numbers of images will also be limited by factors like context window size. As a result, the performance gains from MIA-DPO will gradually diminish with an increasing number of images. These findings show potential for future research on long-context abilities, such as ROPE extrapolation on LVLMs.
>
> > Q6: Can experiments be conducted on more advanced open-source large vision-language models or those explicitly supporting multi-images, such as the Qwen-VL series and Tarsier?
>
> A6: The Qwen-VL series and Tarsier are both highly valuable works in the field, and we have added the discussion with them in the related work section. However, given the limited time of the rebuttal period, we plan to include the results of combining these models with MIA-DPO in future work. To validate the performance of MIA-DPO on advanced open-source large vision-language models, we primarily used the IXC2.5 model, which supports multi-image and video processing. Applying MIA-DPO to this model demonstrates the robustness of our method on advanced models.

---

> ### Author Response · Authors · 2024-11-24
>
> Dear Reviewer ZLYy,
>
> We deeply appreciate your efforts in reviewing our paper, and we are particularly grateful for the invaluable suggestions that have greatly enhanced its quality.
>
> In response to your suggestions, we first explored the comparison between our method and other multi-image SFT methods (Q1) and conducted ablation experiments by converting MIA-DPO data into SFT data (Q2). Additionally, we provided results on more benchmarks (Q3). Subsequently, we constructed a multi-image VQA test set and performed tests (Q4). Furthermore, we examined the model's performance as the number of images increased (Q5) and discussed its performance in advanced models(Q6).
>
> Could you please read our rebuttal and consider adjusting the score in light of our response? Thank you for your time and effort!
>
> Best,
>
> Authors

---

> ### Author Response · Authors · 2024-11-26
>
> Dear Reviewer ZLYy,
>
> Thank you again for your time and valuable feedback on our paper. We have carefully addressed all your comments in our rebuttal and would greatly appreciate it if you could review our response and consider adjusting your score accordingly.
>
> Your support means a lot to us, and we are grateful for your consideration.
>
> Best regards,
>
> Authors

---

> ### Author Response · Authors · 2024-11-27
>
> Dear Reviewer ZLYy,
>
> I hope this message finds you well. I’m writing to kindly follow up on the review process for our paper. We deeply value your expertise and would greatly appreciate it if you could provide your feedback and scoring at your earliest convenience to help progress the review.
>
> We humbly hope for your positive consideration and support in evaluating our work. Your recognition of the effort we’ve put into this research would mean a great deal to us.
>
> Thank you so much for your time and thoughtful review.
>
> Best regards,
>
> Authors

---

> ### Author Response · Authors · 2024-11-28
>
> Dear Reviewer ZLYy,
>
> I hope this message finds you well. As we are in the final stages, we would greatly appreciate your valuable feedback and scoring at your earliest convenience.
>
> We sincerely understand how busy you must be, and we are deeply grateful for the time and effort you’ve already dedicated to reviewing our paper. Your support and thoughtful feedback are extremely important to us, and we would be thankful for any comments you can provide.
>
> Thank you once again for your time and consideration.
>
> Best regards,
>
> Authors

---

> ### Author Response · Authors · 2024-12-01
>
> Dear Reviewer ZLYy.
>
> The deadline is approaching. Now two reviewers post their response to our rebuttal, and our paper received mixed results (one 6 rating and one 5 rating). We have not received any response from you. Can you read our rebuttal and post your comments? Thanks.

---

> ### Author Response · Authors · 2024-12-02
>
> Dear Reviewer ZLYy,
>
> I hope this message finds you well. As the deadline is fast approaching, I would like to kindly inquire about the status of your feedback on our rebuttal. We have received responses from three reviewers, with mixed ratings (two 6 ratings and one 5 rating), but we have not yet received any comments from you. If possible, could you kindly review our rebuttal and provide your feedback at your earliest convenience?
>
> Thank you very much for your time and consideration.
>
> Best regards,
>
> Authors

---

> > ### Comment · Reviewer_ZLYy · 2024-12-02
> >
> > Thank you to the author for the detailed experiments, which have partially addressed my concerns. I have carefully read the comments from other reviewers, and I have decided to **maintain my score**. I believe this paper falls slightly below the acceptance standard for ICLR. My points are:
> >
> > 1. Essentially, this paper raises and solves a problem based on a baseline where multi-image pretraining is **very insufficient**. However, it’s possible that this problem might not exist with a baseline where multi-image pretraining is sufficient.
> >
> > 2. NLL is indeed a common technique to compensate for the shortcomings of DPO, but the performance improvement it brings on multi-image metrics clearly indicates that a certain degree of **direct modeling (e.g. SFT) of multi-images is effective**, and may even be more effective than your DPO-based method.

---

> > > ### Author Response · Authors · 2024-12-02
> > >
> > > Dear Reviewer ZLYy,
> > >
> > > > Q1: Essentially, this paper raises and solves a problem based on a baseline where multi-image pretraining is very insufficient. However, it’s possible that this problem might not exist with a baseline where multi-image pretraining is sufficient.
> > >
> > > A1: In our experiments, we used two models: LLaVA and the advanced IXC2.5. The latter was **pre-trained on a substantial amount of multi-image data** (e.g., MMDU, ShareGPT4Video, ActivityNet). Additionally, IXC2.5 has been evaluated on multiple multi-image benchmarks. Therefore, we believe for multi-image pre-training models like IXC2.5, we have demonstrated that our method still enhances the model's multi-image understanding performance.
> > >
> > > > Q2: NLL is indeed a common technique to compensate for the shortcomings of DPO, but the performance improvement it brings on multi-image metrics clearly indicates that a certain degree of direct modeling (e.g. SFT) of multi-images is effective, and may even be more effective than your DPO-based method.
> > >
> > > A2:
> > > Here we summarize Table 9 and Table 10 of our paper:
> > >
> > > | Method    | Average of 5 Multi-Image Benchmarks | Average of 7 Single-Image Benchmarks |
> > > | -------- | ------- |------- |
> > > | LLaVA 1.5 | 40.4 | 51.6 |
> > > | $L_{NLL}$ only  |  42.1 (+1.7%)  |  49.9 (-1.7%)  |
> > > | $L_{DPO}$ + $L_{NLL}$ |  43.4  (+3.0%) |  51.7  (+0.1%) |
> > >
> > > - Our DPO method achieves greater improvements (+3.0%) in multi-image performance compared to the SFT method (+1.7%).
> > >
> > > - Thanks to the use of KL divergence and rejected data of $DPO$, our method **maintains stable single-image performance** (+0.1%), whereas **the SFT method may lead to significant drops** in single-image benchmark performance (-1.7%).
> > >
> > > Both of these aspects highlight the advantages of our method over the $\mathcal{NLL}$ only baseline.
> > >
> > > You claim that `SFT may even be more effective than your DPO-based method.` We think your assumption that 'may better' is  not rigorous, just a hypothesis, without evidence, and not objective. We have provided our experimental results in Table 9 and Table 10 to demonstrate that our method performs better than the $\mathcal{NLL}$ only baseline. Whether you believe our results or not, we will open-source all our training code, data, and model weights as our evidence. We think that the reviewer should avoid just saying 'may better' as an excuse. We cannot judge the merits of one paper solely on 'may be better' or 'may be worse' assumptions.
> > >
> > > Best regards,
> > >
> > > Authors

---

> > > > ### Comment · Reviewer_ZLYy · 2024-12-02
> > > >
> > > > I understand your desire to defend your work, and I apologize for any imprecise language I may have used. Allow me to clarify my perspective:
> > > >
> > > > 1. The “sufficient” pretraining of IXC-2.5-7B seems limited, primarily using two to three types of multi-image tasks (mainly video-based) with restricted data (if the full sharegpt4video dataset was used, given its quality, I wish it good luck). This may significantly hinder its performance in multi-image and video understanding compared to Qwen series and Tarsier. It’s not widely recognized as a baseline in the community. Notably, **many reported metrics for IXC-2.5-7B appear lower than in the original paper.** Could you explain this discrepancy?
> > > >
> > > > 2. While acknowledging DPO’s role, I **question its potential** for several reasons:
> > > > - Tables 11 and 12 show limited improvement over SFT on MMDU and Mantis (potentially within margin of error, based on my understanding of these benchmarks).
> > > > - Table 12 demonstrates constrained performance gains on the 13B model.
> > > > - Table 19 indicates diminishing improvement in *true* multi-image scenarios.
> > > > - Tables 2 and 17 reveal inevitable performance degradation in single-image scenarios.
> > > >
> > > >   Given the manual and complex nature of MIA-DPO training, would a method like COSA [1] be more efficient and scalable, improving performance during pretraining at minimal cost?
> > > >
> > > > Overall, I don’t deny MIA-DPO’s value or impact. However, I question whether its contribution aligns with the prestige of an ICLR publication.
> > > >
> > > > [1] COSA: Concatenated Sample Pretrained Vision-Language Foundation Model

---

> > > > > ### Author Response · Authors · 2024-12-02
> > > > > **Response to Reviewer ZLYy (1/2)**
> > > > >
> > > > > > I apologize for any imprecise language I may have used
> > > > >
> > > > > Indeed, some of your previous assumptions were not rigorous enough. We are glad that some of your concerns in the previous discussion have been addressed. Here we discuss your remaining concerns and we believe you will present rigorous and fair comments :)
> > > > >
> > > > > > Q1-1: The “sufficient” pretraining of IXC-2.5-7B seems limited, primarily using two to three types of multi-image tasks (mainly video-based) with restricted data (if the full sharegpt4video dataset was used, given its quality, I wish it good luck). This may significantly hinder its performance in multi-image and video understanding compared to Qwen series and Tarsier. It’s not widely recognized as a baseline in the community.
> > > > >
> > > > > A1-1: You strongly criticized two previously published papers (sharegpt4video, InternLM-XC2.5). Since the discussion of the advantages or drawbacks of these two papers is irrelevant to our paper, here we just clarify some factual errors and discuss why we chose LLaVA/IXC as our baselines to compare with:
> > > > >
> > > > > - According to the paper of InternLM-XC2.5, we find that they use diverse multi-image/video datasets, including document understanding (e.g., DUDE), multi-turn and multi-image dialog understanding (e.g., MMDU), and video understanding (e.g., sharegpt4video).
> > > > >
> > > > > - You strongly criticized the quality of one published paper (sharegpt4video) without any evidence. We argue that the advantages or drawbacks of sharegpt4video are irrelevant to our paper: we don't know whether your assumption is correct or not and if the full sharegpt4video dataset is used. Again, sharegpt4video is irrelevant to our paper: the Sharegpt4video focuses on video captioning in pre-training while our MIA-DPO is applied to the post-training (RLHF) stage.
> > > > >
> > > > > - We chose the recent InternLM-XC2.5 as our baseline to prove that MIA-DPO is agnostic to different LVLMs ( LLaVA, InternLM-XC2.5). We think LLaVA (accepted to NeurIPS 2023) and IXC (accepted to NeurIPS 2024) are valuable baselines and have been accepted by the community.
> > > > >
> > > > > - We have added the discussion with QWen and Tarsier in the related work section. The release time of QWen2-VL's technical report and ICLR 2025 submission deadline is less than one month, so we did not report the results of QWen2-VL. Given the limited time of the rebuttal period, we plan to include the results of combining these models with MIA-DPO in future work.
> > > > >
> > > > >
> > > > > > Q1-2: Notably, many reported metrics for IXC-2.5-7B appear lower than in the original paper. Could you explain this discrepancy?
> > > > >
> > > > > A1-2: This is not true. We evaluated 12 benchmarks (5 multi-image, 7 single-image benchmarks), and on 11 of 12 benchmarks the difference between our reported results with the IXC-2.5-7B is less than 0.5. We evaluated both the official IXC-2.5-7B checkpoint and our method in the same environment to ensure a fair comparison, and the discrepancies with the original paper fall within a normal range (e.g., MMStar 0.2, MMBench 0.3, Math 0.5). One exception is the MVBench. Our paper used a consistent frame rate of 4 frames, in line with the LLaVA model, while the original paper used a higher frame rate, which accounts for the difference.
> > > > >
> > > > > > Q2-1:Tables 11 and 12 show limited improvement over SFT on MMDU and Mantis (potentially within margin of error, based on my understanding of these benchmarks)
> > > > >
> > > > > A2-1: This is not true. First, our method achieves a clear improvement over the baseline (LLaVA +3.0%, IXC2.5 +4.3%) while maintaining the original single-image performance(LLaVA 51.6, MIA-DPO 51.7, +0.1\% in Table 10). Regarding the comparison with Mantis and MMDU data, it is important to note that both of these methods use a large amount of additional multi-image data (721k for Mantis, 45k for MMDU) as well as the expensive QAs labeled by GPT-4, whereas our method only synthetic 27k data from LLaVA 665K in a low-cost manner. By using our low-cost and easy-to-scale data pipeline, we have achieved better performance, which we believe demonstrates a significant advantage of our method in terms of data efficiency.
> > > > >
> > > > > > Q2-2:Table 19 indicates diminishing improvement in true multi-image scenarios.
> > > > >
> > > > > A2-2: This is not true. The main focus of the experiment in Table 19 is testing how the model's performance changes as the number of images increases. The data in the table shows that as the number of images increases, the model's ability to handle multi-image tasks decreases, but still consistently boosts the performance of the baseline. We think Table 19 does not reflect a diminishing improvement in true multi-image scenarios. On the contrary, all the multi-image benchmarks we tested are true multi-image scenarios, and our method achieved performance improvements on these benchmarks  (LLaVA +3.0%, IXC2.5 +4.3%).

---

> ### Author Response · Authors · 2024-12-02
> **Response to Reviewer ZLYy (2/2)**
>
> >  Q2-3: Tables 2 and 17 reveal inevitable performance degradation in single-image scenarios.
>
> A2-3: This is not true. Due to the model being trained with DPO on multi-image data, there are naturally some fluctuations in single-image benchmark performance, with some metrics improving and others declining. However, when considering the overall performance across multiple single-image benchmarks, the model's single-image performance remains at its original level, and shows a significant advantage compared to the results obtained using only NLL loss.(LLaVA 51.6, LLAVA+NLL 49.9, -1.7\%, MIA-DPO 51.7, +0.1\% in Table 10). Should we consider the average results on the 8 datasets, or just look at the decreasing result on one dataset and ignore the other seven?
>
> > Q2-4: Given the manual and complex nature of MIA-DPO training, would a method like COSA [1] be more efficient and scalable, improving performance during pretraining at minimal cost?
>
> A2-4: Our method is completely different from COSA [1], so we believe there is no meaningful basis for comparison between the two. COSA is designed for video description, while our method focuses on achieving model alignment through DPO. COSA is a pre-training strategy, whereas we focus on post-training. COSA primarily proposes a new video understanding model, whereas our method aims to mitigate multi-image hallucinations and achieve model alignment.
>
> [1] COSA: Concatenated Sample Pretrained Vision-Language Foundation Model

---

> ### Author Response · Authors · 2024-12-03
>
> Dear Reviewer ZLYy,
>
> We are sorry that we have not received a response from you since we last response to your questions. Do you have any further concerns? We will do our utmost to address them. Given the mixed results with two ratings of 6 and one of 5, we would be grateful if you could clarify any remaining concerns that we may not have fully addressed. We truly look forward to your response. Thank you again for your time and thoughtful feedback.
>
> With sincere appreciation,
>
> The Authors

---

### Official Review · Reviewer_mzdh · 2024-11-04

**Soundness:** 3
**Presentation:** 3
**Contribution:** 3
**Rating:** 6
**Confidence:** 3

**Summary:**

This paper introduces MIA-DPO, a visual preference alignment approach for LVLMs that handles multi-image tasks. The authors propose extending single-image data with unrelated images and using attention patterns to detect hallucinations, allowing construction of chosen/rejected pairs without human annotation or external APIs. The method shows promising results on both multi-image and single-image benchmarks.
Overall, I believe this is a solid paper that addresses an important and under-explored problem. It makes meaningful contributions to visual preference alignment for multi-image scenarios, though there are some aspects that need clarification.

**Strengths:**

* While there's been significant work on visual preference alignment for single images, multi-image alignment remains largely unexplored despite its real-world importance. The authors identify key challenges like data scarcity and high annotation costs.
* Rather than collecting new multi-image data (which would be expensive), they augment existing single-image data through sequence, grid collage, and pic-in-pic formats. Using attention patterns to select rejected samples is an elegant way to avoid costly human annotation or API calls.
* The authors test on 5 multi-image benchmarks and 7 single-image ones, showing clear improvements (+3.0% on LLaVA-1.5, +4.3% on InternLM-XC2.5) while maintaining single-image performance. The ablations examining different components are thorough.

**Weaknesses:**

However, I have several concerns that I think should be addressed:
My main concern is about the attention-based selection method. While using attention patterns to detect hallucinations is interesting, the threshold selection feels somewhat ad-hoc. The authors set different thresholds based on image counts (0.7/0.6/0.5/0.5 for 2/3/4/5 images), but there's limited justification for these specific values. I wonder if these thresholds could be learned automatically rather than manually set.
The relationship between augmentation strategies and model performance isn't fully explored. While the authors try three formats (sequence, grid collage, pic-in-pic), there's limited analysis of how different ratios of these formats affect results. Some discussion of which formats work better for what types of queries would be valuable.
The ablations, while informative, don't completely isolate the impact of different components. It would be interesting to see how model size affects the attention patterns and overall performance. Additionally, some analysis of failure cases would help understand the method's limitations.

**Questions:**

Have you analyzed how sensitive the results are to the attention thresholds? What happens if you use uniform thresholds across different image counts?
For the post-selection filtering, how did you choose the specific metrics (perplexity, length ratio, edit distance)? Were other metrics considered?
Your method improves most on the Mantis benchmark (+11.1%). Any insights into why the gains are particularly large there?

---

> ### Author Response · Authors · 2024-11-21
> **Authors Rebuttal to Reviewer mzdh (1/2)**
>
> Dear Reviewer mzdh,
>
> We sincerely thank you for your thorough feedback: identifying the key challenges, avoiding costly human annotation or API calls, clear improvements, and thorough ablations. We address your questions below and have incorporated all feedback in the revised version. ***All new material added to the revised manuscript has been highlighted in red text for better visibility.***
>
>
> > Q1-1: My main concern is about the attention-based selection method. While using attention patterns to detect hallucinations is interesting, the threshold selection feels somewhat ad-hoc. The authors set different thresholds based on image counts (0.7/0.6/0.5/0.5 for 2/3/4/5 images), but there's limited justification for these specific values. I wonder if these thresholds could be learned automatically rather than manually set.
>
> A1-1: Thank you for your valuable feedback. Our attention-based selection method is inspired by our observation that attention distributions become more dispersed as the number of images increases. This necessitates the use of different thresholds to effectively identify relevant information. While we acknowledge the potential benefits of automatic threshold learning, our current approach is based on statistical analysis (see Fig. 5): before setting the threshold, we analyze the attention ratio distributions for 1k samples of each of the three data formats. Based on these distributions, we set the threshold to retain 50%-70% of the samples. Furthermore, our experiments demonstrate that our MIA-DPO is relatively robust to different threshold ranges (see A1-2). We agree to the value of exploring automated techniques to optimize threshold selection and will consider this as a promising direction for future research.
>
> > Q1-2: Have you analyzed how sensitive the results are to the attention thresholds?
>
> A1-2: Thanks to your suggestion, we have conducted multiple ablation experiments with different threshold ranges, see Tab. 13 and Tab. 14. We observe that our MIA-DPO is relatively robust to different threshold ranges, and our default choices (0.7/0.6/0.5/0.5) performs slightly better than other choices.
>
> > Q1-3: What happens if you use uniform thresholds across different image counts?
>
> A1-3: We have added the baseline that uses uniform thresholds, and results are shown in the fourth row of Tab. 13 and Tab. 14. The experimental results demonstrate that using a uniform threshold can negatively impact performance. This observation is due to the attention ratio distributions varying significantly depending on the number of images (see Fig. 5), which indicates that a one-size-fits-all threshold is not optimal. By adjusting the threshold based on the number of images, we can improve the model's ability to generate accurate and coherent responses.
>
> > Q2: The relationship between augmentation strategies and model performance isn't fully explored. While the authors try three formats (sequence, grid collage, pic-in-pic), there's limited analysis of how different ratios of these formats affect results. Some discussion of which formats work better for what types of queries would be valuable.
>
> A2:  Valuable feedback! First, we conducted ablation experiments for each data type, as presented in Table 3 of the paper. We explored the effects of using each data type individually, and the results show that the performance improvements from using each data type alone are similar but do not surpass the case where all three types of data are combined.
> Additionally, based on your insightful suggestion, we have included ablation studies on different data proportions in Tab. 13 and Tab. 14. We adjust the proportions of the three data types and evaluate the results. The results indicate that the model's performance on both multi-image and single-image tasks remains stable, with only minor fluctuations. This demonstrates the robustness of our model across different data proportion settings. In conclusion, none of the data types showed a particularly significant advantage over others. When used in combination, the three data types achieve better performance than any single type used individually.
>
> > Q3: The ablations, while informative, don't completely isolate the impact of different components.
>
> A3: Thanks for your suggestion. In Tab. 15, we have conducted ablation experiments on different components of post-selection. Specifically, we individually used perplexity (ppl), length ratio, and edit distance as the sole components of post-selection for the experiments. The experiments show that the performance of using each component individually is similar, but the combined use of all three methods yields the best results.

---

> ### Author Response · Authors · 2024-11-21
> **Authors Rebuttal to Reviewer mzdh (2/2)**
>
> > Q4: It would be interesting to see how model size affects the attention patterns and overall performance.
>
> A4: Good question! To explore the effectiveness of larger models, we apply the MIA-DPO with the LLaVa-v1.5-13B model. We observe that the attention distribution patterns of the 13B model are largely consistent with those of the 7B model. We have included the results of MIA-DPO + LLaVa-v1.5-13B on multi-image and single-image benchmarks in Tab. 16 and Tab. 17. These results further validate our conclusions about the effectiveness of MIA-DPO, demonstrating that it can consistently improve the performance of larger size models.
>
> > Q5: Additionally, some analysis of failure cases would help understand the method's limitations.
>
> A5: We have provided some erroneous cases in Fig. 11.  The cases presented are from the multi-image QA of the MMMU benchmark. From the model's responses, we can observe that although MIA-DPO has improved the model's multi-image understanding and reasoning capabilities, the model may still make errors when encountering questions from out-of-domain knowledge (e.g., fine-grained plant classification, medical image processing), which is not present in our training data. We believe extending the training data to more diverse domains will alleviate these failure cases.
>
> > Q6: For the post-selection filtering, how did you choose the specific metrics (perplexity, length ratio, edit distance)? Were other metrics considered?
>
> A6: The use of perplexity is based on the observation that when the model outputs a high perplexity, it indicates a lack of confidence in its responses, making it more prone to generating hallucinations. The length ratio is employed to ensure that there is no significant disparity in length between the chosen and rejected samples, as such differences could negatively affect the optimization of the loss function. Edit distance is used to prevent the chosen and rejected samples from being too similar, as overly similar samples (e.g., "apple" and "apples") could hinder the model’s ability to learn meaningful patterns. While our selected metrics have proven effective, other techniques, such as semantic similarity  metrics, could be explored to further refine the post-selection process. These additional metrics can help ensure the diversity of the selected data.
>
> > Q7: Your method improves most on the Mantis benchmark (+11.1%). Any insights into why the gains are particularly large there?
>
> A7: Good question! The IXC2.5 model itself is relatively advanced with strong learning capabilities, which may explain the substantial performance gains achieved after applying MIA-DPO. On the other hand, we find that the original IXC2.5 model lacks instruction-following abilities when answering questions in the Mantis benchmark. This often resulted in the model providing lengthy descriptions instead of concise answers for multiple-choice questions as shown in Fig. 12. After applying MIA-DPO, the model prefers to answer the question that meets the format requirement, which contributed to its higher scores on Mantis.

---

> ### Author Response · Authors · 2024-11-24
>
> Dear Reviewer mzdh,
>
> We deeply appreciate your efforts in reviewing our paper, and we are particularly grateful for the invaluable suggestions that have greatly enhanced its quality.
>
> In response to your suggestions, we conducted ablation experiments on both different and unified attention thresholds(Q1). Additionally, we further explored augmentation strategies, examining the impact of different data ratios on the model(Q2). We also broke down the different components for more in-depth ablation studies(Q3). Subsequently, we tested the effect of model size on our method(Q4) and presented some failure cases(Q5). Finally, we analyzed the post-selection process and potential alternative approaches(Q6), as well as the significant performance improvements observed on the Mantis(Q7).
>
> Could you please read our rebuttal and consider adjusting the score in light of our response? Thank you for your time and effort!
>
> Best,
>
> Authors

---

> ### Author Response · Authors · 2024-11-26
>
> Dear Reviewer mzdh,
>
> Thank you again for your time and valuable feedback on our paper. We have carefully addressed all your comments in our rebuttal and would greatly appreciate it if you could review our response and consider adjusting your score accordingly.
>
> Your support means a lot to us, and we are grateful for your consideration.
>
> Best regards,
>
> Authors

---

> ### Author Response · Authors · 2024-11-27
>
> Dear Reviewer mzdh,
>
> I hope this message finds you well. I’m writing to kindly follow up on the review process for our paper. We deeply value your expertise and would greatly appreciate it if you could provide your feedback and scoring at your earliest convenience to help progress the review.
>
> We humbly hope for your positive consideration and support in evaluating our work. Your recognition of the effort we’ve put into this research would mean a great deal to us.
>
> Thank you so much for your time and thoughtful review.
>
> Best regards,
>
> Authors

---

> ### Author Response · Authors · 2024-11-28
>
> Dear Reviewer mzdh,
>
> I hope this message finds you well. As we are in the final stages, we would greatly appreciate your valuable feedback and scoring at your earliest convenience.
>
> We sincerely understand how busy you must be, and we are deeply grateful for the time and effort you’ve already dedicated to reviewing our paper. Your support and thoughtful feedback are extremely important to us, and we would be thankful for any comments you can provide.
>
> Thank you once again for your time and consideration.
>
> Best regards,
>
> Authors

---

> ### Comment · Reviewer_mzdh · 2024-11-29
>
> Thank the authors for addressing all my concerns, I'll keep my score.

---

> > ### Author Response · Authors · 2024-11-30
> >
> > Dear Reviewer mzdh,
> >
> > Thank you very much for your response and for maintaining your original score 6. We are truly grateful for your recognition of the quality and significance of our work. Your feedback and encouragement mean a great deal to us and motivate us to continue improving our research.
> >
> > Sincerely,
> >
> > Authors

---

### Official Review · Reviewer_ePWn · 2024-11-04

**Soundness:** 3
**Presentation:** 3
**Contribution:** 2
**Rating:** 5
**Confidence:** 4

**Summary:**

The paper presents MIA-DPO, a multi-image DPO dataset construction approach for LVLMs without annotations from human or external models. The paper collects prompts from existing single-image dataset like LLaVA 665k, augmenting each data with unrelated images, which derives synthetic prompts of multiple images. Taking advantage of transformer's attention scores, rejected samples can be automatically filtered out to build pairwise preference answers for DPO. In addition, some post-selection methods are proposed for data cleaning. Experiment shows improvements on most multi-image and single-image benchmarks, and the capability to overcome hallucinations duo to the interference from unrelated images in prompts.

**Strengths:**

- The paper is clearly written. Most details are provided, so it should be easy to reimplement.

- The proposed method is well-motivated. Hallucination suppression is an important topic in LVLMs. The paper presents a straight-forward way to address hallucinations from unrelated images in multi-image scenario, which seems very easy to follow and does not rely on human annotations.

- Experiments are well organized. Extensive evaluations show the effectiveness of the proposed method. An interesting phenomenon is, although the synthetic DPO prompts appear highly biased, improvements are still obtained on **general** single-image and multi-image evaluations.

**Weaknesses:**

## Significance

My biggest concern is, the proposed synthetic method is designed to deal with the interference from unrelated images, however, which seems only a **corner case** in LVLMs. In practice, typically all images in the question prompt contribute to the final answer. It is a great challenge (but more important) to generate preference data for such *real* multi-image prompt at scale without much cost. The paper does not involve the topic. Therefore, I think the title "Multi-Image Augmented Direct Preference Optimization" may somewhat overclaim the contribution.

To improve the work, I think the authors may defend their contributions by analyzing the root of the improvements on general benchmarks. For example, an interesting topic may be "does the proposed synthetic DPO dataset also help to suppress more general hallucinations other than those in Line 212?"; if yes, why?


## Experiment

I feel some empirical results are weak because:

- According to Table 3, parts of the improvements are resulted from data cleaning (especially on MMMU and BLINK), which is not the unique technical contribution of the paper.

- In Eq 5, the NLL term plays a role of supervised fine-tuning with only chosen answers. It is an important baseline (i.e. fine-tuning with isolated NLL term) which can reflect the effectiveness of rejected samples, but not provided in the experiment.

**Questions:**

* In Eq 4, please clarify how to calculate R(y) in details. Is the summation of attention values computed over all layers, attention heads and tokens subject to the given image/region?

---

> ### Author Response · Authors · 2024-11-21
> **Authors Rebuttal to Reviewer ePWn (1/2)**
>
> Dear Reviewer ePWn,
>
> We sincerely thank you for your thorough feedback: the paper is clearly written, the proposed method is well-motivated, easy to reimplement, and the experiments are well organized. We address your questions below and have incorporated all feedback in the revised version. ***All new material added to the revised manuscript has been highlighted in red text for better visibility.***
>
> > Q1:My biggest concern is, the proposed synthetic method is designed to deal with the interference from unrelated images, however, which seems only a corner case in LVLMs. In practice, typically all images in the question prompt contribute to the final answer. It is a great challenge (but more important) to generate preference data for such real multi-image prompt at scale without much cost. The paper does not involve the topic. Therefore, I think the title "Multi-Image Augmented Direct Preference Optimization" may somewhat overclaim the contribution.
>
> A1: We agree that our synthetic method indeed only requires considering parts of (rather than all) images to infer the final answer. Despite its limitations, our MIA-DPO approach has demonstrated significant improvements on general single-image and multi-image benchmarks. This suggests that our method can effectively address general multi-image hallucination issues even when not explicitly designed for the most complex scenarios. And we will present reasonable explanations in A2.
> We acknowledge your suggestion that generating preference data that requires considering all input images is an important direction for future research. However, this is a significant challenge, requiring strong models (e.g., GPT-4o) or human expertise as the annotator, which is both costly and time-consuming. We are committed to exploring more sophisticated techniques (e.g., incorporating domain-specific knowledge to generate more accurate and informative preference data) to handle complex multi-image prompts in future work.
> Although without involving the complex multi-image prompts, we believe that our work represents a valuable first step and provides a strong foundation for future research in multi-image preference alignment.
>
> > Q2: To improve the work, I think the authors may defend their contributions by analyzing the root of the improvements on general benchmarks. For example, an interesting topic may be "does the proposed synthetic DPO dataset also help to suppress more general hallucinations other than those in Line 212?"; if yes, why?
>
> A2: Thank you for your valuable feedback! Here we provide more detailed explanations about why MIA-DPO enhances model performance on general benchmarks and suppresses more general multi-image hallucinations.
> About Training Paradigm:
> - MIA-DPO's training process necessitates the processing of a significantly larger number of image tokens, improving the model's ability to take multiple images as inputs.
> - Our careful selection of chosen and rejected samples in the DPO framework further enhances the model's ability to discern subtle differences and nuances between images.
> About Hallucination Suppression:
> - Our synthetic DPO data is meticulously designed to address a diverse range of multi-image hallucination scenarios, including those involving unrelated images, conflicting information, and ambiguous prompts. By exposing the model to these challenging examples, MIA-DPO effectively learns to identify and mitigate such hallucinations.
> In conclusion, MIA-DPO's unique training paradigm and  data generation strategy empower models to handle complex multi-image prompts to some extent, ultimately leading to more robust and accurate models.
>
> > Q3: According to Table 3, parts of the improvements are resulted from data cleaning (especially on MMMU and BLINK), which is not the unique technical contribution of the paper.
>
> A3: The post-selection procedure, which involves removing outliers and low-quality data, is an integral part of our method. This step, combined with our multi-image prompt construction and attention-aware selection techniques, leads to a substantial improvement in the overall quality of the DPO dataset. From an average perspective, the primary driver of our method's superior performance on multi-image benchmarks (Table 3) is our multi-image prompt construction and attention-aware selection techniques, rather than data cleaning alone.

---

> ### Author Response · Authors · 2024-11-21
> **Authors Rebuttal to Reviewer ePWn (2/2)**
>
> > Q4: In Eq 5, the NLL term plays a role of supervised fine-tuning with only chosen answers. It is an important baseline (i.e. fine-tuning with isolated NLL term) which can reflect the effectiveness of rejected samples, but not provided in the experiment.
>
> A4: Excellent suggestion!  We have incorporated a new baseline (fine-tuning with only the NLL term). Results are presented in Tables 9 and 10 for multi-image and single-image benchmarks, respectively.
> * On multi-image benchmarks (Table 9), fine-tuning with the NLL term (second row) yields a performance improvement over the LLaVA-1.5 baseline. However, MIA-DPO (third row) consistently outperforms the NLL-only baseline, demonstrating the significant contribution of negative samples to model improvement.
> * On single-image benchmarks (Table 10), the NLL-only baseline (second row) leads to a performance degradation compared to LLaVA-1.5, highlighting the potential risks of incorporating multi-image data during SFT may adversely affect performance on single-image tasks. By contrast, MIA-DPO (third row) maintains performance parity with LLaVA-1.5, thanks to the KL-divergence loss constraint in Eq. (3). This further demonstrates the advantages of MIA-DPO over the NLL-only baseline.
>
> > Q5: In Eq 4, please clarify how to calculate R(y) in detail. Is the summation of attention values computed over all layers, attention heads and tokens subject to the given image/region?
>
> A5: Apologies for not clearly explaining the calculation of R(y). First, we used the attention from the middle layer as the experimental subject (since the phenomenon is most evident in the middle layer). For sequence data, we calculate the total attention value of the image pointed to by the instruction as A_target, and the total attention value of all images as A_sum. For grid collage and pic-in-pic data, we calculate the total attention value of the region in the image pointed to by the instruction as A_target, and the total attention value of the entire image as A_sum. The ratio of A_targe

---

> ### Author Response · Authors · 2024-11-24
>
> Dear Reviewer ePWn,
>
> We deeply appreciate your efforts in reviewing our paper, and we are particularly grateful for the invaluable suggestions that have greatly enhanced its quality.
>
> In response to your suggestions, we first analyzed the issue of related images in Q1, then discussed the root of the improvements and the data cleaning problem in Q2 and Q3, respectively. Additionally, in Q4, we provided ablation experiments on NLL, and in Q5, we provided a detailed explanation of the calculation method for R(y).
>
> Could you please read our rebuttal and consider adjusting the score in light of our response? Thank you for your time and effort!
>
> Best,
>
> Authors

---

> ### Author Response · Authors · 2024-11-26
>
> Dear Reviewer ePWn,
>
> Thank you again for your time and valuable feedback on our paper. We have carefully addressed all your comments in our rebuttal and would greatly appreciate it if you could review our response and consider adjusting your score accordingly.
>
> Your support means a lot to us, and we are grateful for your consideration.
>
> Best regards,
>
> Authors

---

> ### Author Response · Authors · 2024-11-27
>
> Dear Reviewer ePWn,
>
> I hope this message finds you well. I’m writing to kindly follow up on the review process for our paper. We deeply value your expertise and would greatly appreciate it if you could provide your feedback and scoring at your earliest convenience to help progress the review.
>
> We humbly hope for your positive consideration and support in evaluating our work. Your recognition of the effort we’ve put into this research would mean a great deal to us.
>
> Thank you so much for your time and thoughtful review.
>
> Best regards,
>
> Authors

---

> ### Author Response · Authors · 2024-11-28
>
> Dear Reviewer ePWn,
>
> I hope this message finds you well. As we are in the final stages, we would greatly appreciate your valuable feedback and scoring at your earliest convenience.
>
> We sincerely understand how busy you must be, and we are deeply grateful for the time and effort you’ve already dedicated to reviewing our paper. Your support and thoughtful feedback are extremely important to us, and we would be thankful for any comments you can provide.
>
> Thank you once again for your time and consideration.
>
> Best regards,
>
> Authors

---

> > ### Comment · Reviewer_ePWn · 2024-11-30
> > **Not good enough**
> >
> > Thanks for the authors' feedback. Parts of my concerns, e.g. the calculation details of R(y), have been addressed. However, I still think the contribution of the paper is relatively weak currently, which does not meet the standard of ICLR:
> >
> > - As acknowledged in A1 and A2, the proposed synthetic method only generates "pseudo" multi-image prompts, i.e. only one of the images relates to the final answer. The capability of addressing general multi-image hallucination is indirect, however, according to the experiment, I do not think there is a significant improvement.
> >
> > - In contrast, synthesizing "true" multi-image data is more important. For example, one possible direction could be: first, mining related questions in existing single-image datasets; then, prompting language models to construct new multi-image Q/A pairs from a collage of related single-image Q/As. Since the language model is only required to re-organize the provided prompts & answers, maybe open-source models are good enough.
> >
> > - New empirical results are a bit disappointing. For example, in Table 2 the performance drop on OCR bench is significant. And Table 9 shows only using $L_{NLL}$ already leads to most gains (except for NLVR2).

---

> > > ### Author Response · Authors · 2024-12-01
> > > **Rebuttal to ePWn (1/2)**
> > >
> > > Dear Reviewer ePWn,
> > >
> > > We are truly delighted to hear that our revisions have addressed part of your concerns.
> > >
> > > Here we attempt to address the remaining concerns in your response. We believe there are important aspects that we need to clarify further to demonstrate the significance of our approach.
> > >
> > > > Q1: As acknowledged in A1 and A2, the proposed synthetic method only generates "pseudo" multi-image prompts, i.e. only one of the images relates to the final answer. The capability of addressing general multi-image hallucination is indirect, however, according to the experiment, I do not think there is a significant improvement.
> > >
> > > A1:
> > > - Our work achieves multi-image alignment through a low-cost and simple pipeline, achieving performance improvements on several multi-image benchmarks (LLaVA +3.0%, IXC2.5 +4.3%) that we believe is a *significant* performance gain.
> > > - Other reviewers also acknowledge that our proposed method shows **clear** improvements while maintaining single-image performance (reviewer mzdh),  MIA-DPO can **enhance performance** on multi-image tasks such as MVBENCH without compromising single-image metrics like MMMU (reviewer ZLYy), the experiments show the **effectiveness** of the method against the baselines (reviewer DZHj).
> > > - We are surprised that only you claim that `I do not think there is a significant improvement`. Can you further explain what level of improvement would be considered *significant* for you?
> > > - Although our dataset does not include multi-image related QA, as the first implementation of multi-image alignment that is low-cost and easy to realize, the performance on multiple multi-image benchmarks already demonstrates the effectiveness and generalization ability of our method, which is not influenced by 'pseudo' multi-image prompts.
> > >
> > > > Q2: In contrast, synthesizing "true" multi-image data is more important. For example, one possible direction could be: first, mining related questions in existing single-image datasets; then, prompting language models to construct new multi-image Q/A pairs from a collage of related single-image Q/As. Since the language model is only required to re-organize the provided prompts & answers, maybe open-source models are good enough.
> > >
> > > A2: The solution you proposed is indeed an approach worth exploring. The method of using larger models to generate QA is indeed a common approach for generating a lot of data. However, the 7B LVLM models may contain hallucination and miss the image details, and the 70/72B LVLM models still have **inference costs that can not be ignored**.
> > > It is important to note that our method focuses on achieving significant performance improvements with **low cost and low computational requirements**, which we believe is a key advantage of our work, avoiding the expensive API inference costs. Imagine a method with almost zero data construction costs that can achieve performance improvements on multiple multi-image benchmarks — such a method is incredibly cost-effective.
> > >
> > > Many famous works are not perfect at the beginning. For example, the early version of the diffusion model had slow sampling speeds, and a series of fast sampling works based on diffusion modes were developed later to address this issue. It would not be fair to reject the diffusion model just because the early version is not fast enough. Similarly, our method is the *first* to implement multi-image DPO, and we believe there is room for improvement in the future. Based on our approach, you propose an **untested** idea that may further improve our method. But we believe that your untested idea is not a valid reject reason.
> > >
> > > > Q3: New empirical results are a bit disappointing. For example, in Table 2 the performance drop on OCR bench is significant.
> > >
> > > A3:
> > > *Don’t just look at the results of a single dataset and ignore the overall average performance*. The average results of our method on 8 single-image benchmarks remain largely consistent with the original model (LLaVA **51.6**, **MIA-DPO **51.7** in Table 10, +0.1 for MMDU). Compared to the significant performance drop in single-image tasks brought by other multi-image SFT baselines (e.g., -1.7 for NLL-loss only), our method has effectively preserved single-image performance. Since we tested a total of 8 single-image datasets, it is normal to have fluctuations in one dataset. The mean result is more *statistically significant* than the individual result.

---

> > > ### Author Response · Authors · 2024-12-01
> > > **Rebuttal to ePWn (2/2)**
> > >
> > > > Q4: And Table 9 shows only using $L_{NLL}$ already leads to most gains (except for NLVR2).
> > >
> > > A4:
> > > - Simply using NLL loss can indeed bring some performance improvement to the model, but it is important to note that in Table 10, after using only NLL loss, the model's single-image performance significantly decreases (LLaVA 51.6, LLAVA+NLL 49.9, -1.7\%, in Table 10).
> > > - The DPO method, compared to SFT, benefits from the negative data and KL divergence term on policy model $\pi_{\theta}$ in Eq. 3, which not only leads to greater improvements in the model’s multi-image understanding performance (LLaVA +3.0%, IXC2.5 +4.3%) but also preserves the model’s original single-image reasoning ability (LLaVa **51.6**, **MIA-DPO **51.7**, +0.1\%).
> > > - Overall, while the NLL-only baseline can also bring some improvements to multi-image benchmarks, *the extent of the multi-image improvement* and its impact on *single-image ability* are far inferior to our method.
> > >
> > > P.S.: many existing works in the NLP/LLM area have demonstrated that applying RLHF (e.g., using the DPO loss) is better than using the SFT (NLL loss) only, and our observations on multi-image understanding also verify this point. We are confused why you still think merely using SFT loss is enough?

---

> ### Comment · Reviewer_ePWn · 2024-12-01
> **Additional comments**
>
> ### Why I do not think there is a significant improvement
>
> I think the significance of an improvement varies from different fields and methods. For ImageNet classification and COCO detection, each 1% gain could be valuable. But it is LVLMs -- for example,  for 7B model on MMMU, LLaVA-v1.5 is 35.1, InternLM-XC2.5 is 41.4, and recently Qwen2-VL-7B reaches 54.1, which increases 19% within only one year! It is the power of data and engineering. With the rapid increasing of baselines, for a research paper I value technical contributions more than simply benchmark gains (unless the improvement is as much as that of Qwen2-VL). In other words, I do not think the result (itself) in Table 1 and Table 2 is "significant", if the paper does not address a technically impressive problem.
>
> The authors may argue that their method obtains consistently improvements on two base models, which may generalize to more baselines. Yeah, it may be true, however, does not surprise me very much -- since more multi-image prompts (Sec 3.3.1) with proper data cleaning (Sec 3.3.2) are introduced, multi-image benchmarks are expected to improve (Table 1), and some single-image benchmarks drop due to distribution changes (Table 2). But, the key issue is the synthetic method (which I think is the core technical contribution) is too straight forward, only dealing with the simplest case in multi-image QA. I do not think in the paper gives new academic insights, except for a few engineering practice. It does not mean the paper has no contributions, but may not meet the standard of a top conference like ICLR.
>
> ### About my solution
>
> It is an example to explain which problem I think is significant (and timely to the current development of LVLMs), not a reason to reject the paper (sorry for the potential misunderstanding). I mean, if the paper aims to synthetic "true" multi-image data and obtains improvements like those in Table 1, I would definitely vote acceptance. I do not agree it is analogous to the early versions of diffusion models; at least to me, diffusion models have nontrivial technical contributions regardless of some shortcomings.
>
> ### Concerns on $L_{NLL}$
>
> The paper analyzes multi-image hallucinations. In the proposed DPO framework, the negative samples should take responsibility for multi-image hallucination suppression. However, in Table 9, if $L_{NLL}$ individually has already obtained most gains, I am afraid whether the claim is valid.

---

> ### Author Response · Authors · 2024-12-02
> **Rebuttal to Reviewer ePWn (1/3)**
>
> > Q1: I think the significance of an improvement varies from different fields and methods. For ImageNet classification and COCO detection, each 1% gain could be valuable. But it is LVLMs -- for example, for 7B model on MMMU, LLaVA-v1.5 is 35.1, InternLM-XC2.5 is 41.4, and recently Qwen2-VL-7B reaches 54.1, which increases 19% within only one year! It is the power of data and engineering. With the rapid increasing of baselines, for a research paper I value technical contributions more than simply benchmark gains (unless the improvement is as much as that of Qwen2-VL). In other words, I do not think the result (itself) in Table 1 and Table 2 is "significant", if the paper does not address a technically impressive problem.
>
> A1:
> We disagree with the reviewer’s assessment that a 3-4 point improvement is insignificant. While it is true that models like Qwen2-VL-7B have demonstrated remarkable performance gains, this does not weaken the value of our improvements. A 3-4 point improvement on key benchmarks, especially when achieved through a low-cost, simple, and easily realizable pipeline like ours, is not trivial.
>
> **We cannot disguise replacement of the concept. C > A does not mean that A + B > A is not significant.** The QWen2-VL has a different architecture, pre-training/SFT data, and training schedule from our baseline (LLaVA 1.5, IXC 2.5). Our method is general and we promise to further try to apply our method to recent baselines (e.g., QWen2-VL). In addition, we would like to state that LLaVA (accepted to NeurIPS 2023) and IXC (accepted to NeurIPS 2024) are valuable baselines. The release time of QWen2-VL's technical report and ICLR 2025 submission deadline is less than one month, so we did not report the results of QWen2-VL. **Do you think it is reasonable to weaken the contribution of a paper by using a tech report that has not been peer-reviewed and is released in the same period?**
>
> In the field of machine learning, incremental improvements are often the result of highly effective techniques that can be deployed more easily and at a lower cost. It is important to note that these gains, while perhaps smaller than those of Qwen2-VL-7B compared to LLaVA, are still meaningful and demonstrate the practical efficacy of our approach, especially considering the constraints and trade-offs involved.
>
> Other reviewers have clearly acknowledged that our proposed method delivers noticeable improvements while maintaining single-image performance (reviewer mzdh). In particular, MIA-DPO demonstrates clear enhancement in multi-image tasks, without sacrificing single-image metrics like MMMU (reviewer ZLYy). Our experiments also validate the effectiveness of the method, showing strong performance against baselines (reviewer DZHj). In any case, we believe our approach makes a valuable contribution by delivering tangible, measurable improvements.
>
>
> > Q2:The authors may argue that their method obtains consistently improvements on two base models, which may generalize to more baselines. Yeah, it may be true, however, does not surprise me very much -- since more multi-image prompts (Sec 3.3.1) with proper data cleaning (Sec 3.3.2) are introduced, multi-image benchmarks are expected to improve (Table 1), and some single-image benchmarks drop due to distribution changes (Table 2). But, the key issue is the synthetic method (which I think is the core technical contribution) is too straight forward, only dealing with the simplest case in multi-image QA. I do not think in the paper gives new academic insights, except for a few engineering practice. It does not mean the paper has no contributions, but may not meet the standard of a top conference like ICLR.

---

> ### Author Response · Authors · 2024-12-02
> **Rebuttal to Reviewer ePWn (2/3)**
>
> A2:
> Ablation experiments on SFT and data cleaning have already demonstrated that "multi-image prompts (Sec 3.3.1) with proper data cleaning (Sec 3.3.2)" are **not the primary reasons** for the performance improvements achieved by our method. Additionally, the statement "some single-image benchmarks drop" is also inaccurate. As shown in Table 10, our method even achieves an improvement **in average** single-image performance on LLaVA. **Should we consider the average results on the 8 datasets, or just look at the decreasing result on one dataset and ignore the other seven?**
>
> Besides, we think the reviewer mistakes in identifying the academic insights and novelty of a paper. We recommend the reviewers to read [1].
>
> We believe novelty or academic insights should not be limited to designing technical complex algorithms (which looks like the understanding of the reviewer ePWn), because:
>
> - *A comprehensive analysis that is not found by previous papers* can also be worth publishing. We conducted an in-depth analysis of the types of multi-image hallucinations and their differences from single-image hallucinations, and explored the underlying mechanisms of multi-image hallucination generation through attention.
>
> - *A simple and effective approach* can also be worth publishing. Our MIA-DPO is easy to implement, low-cost, and efficient, and can be scaled to different models and larger datasets.
>
> [1] Novelty in Science: A guide for reviewers. Michael J. Black.
>
> > Q3: I mean, if the paper aims to synthetic "true" multi-image data and obtains improvements like those in Table 1, I would definitely vote acceptance. I do not agree it is analogous to the early versions of diffusion models; at least to me, diffusion models have nontrivial technical contributions regardless of some shortcomings.
>
> A3:
>
> We would like to reiterate that our method is not limited by the lack of "true" multi-image data; rather, it achieves significant performance improvements through an efficient and low-cost pipeline. It is a **novel approach** to multi-image alignment that achieves **significant improvements** without incurring the high computational costs associated with large language models (LVLMs). The essence of our method lies in its **low-cost, low-computation design**, which provides an effective solution for multi-image tasks without the need for expensive inference or data construction.
>
> While we acknowledge that generating true multi-image data could be an interesting direction, we believe it is important to recognize the practicality and impact of our current approach. Synthesizing multi-image data using large models such as 7B or 70/72B LVLMs, as you suggested, may indeed be useful, but it **comes with significant inference costs** and potential **hallucinations** from the models, which could diminish the reliability and accuracy of the generated data. Moreover, such approaches would likely **increase the complexity and cost** of the system, which runs counter to the advantages we emphasize in our work—**simplicity and cost-effectiveness**.
>
> We understand your point, but we would like to clarify that our analogy to the early versions of diffusion models was not intended to make an in-depth comparison between the two methods. Rather, it was simply to illustrate a broader principle: **many influential works are not perfect in the beginning**, but that does not diminish their significance or the value they provide as starting points for future improvements.
>
> We believe the core message here is that **innovations often start with simple, effective, and easy-to-follow solutions**, and that does not make them any less important. By dismissing our approach simply because it may be optimized further in the future, we risk overlooking the current value it already provides. **Can we weaken the contribution of one paper simply because it may be optimized further in the future (but not verified)?**

---

> ### Author Response · Authors · 2024-12-02
> **Rebuttal to Reviewer ePWn (3/3)**
>
> > Q4: Concern on NLL... In the proposed DPO framework, the negative samples should take responsibility for multi-image hallucination suppression. However, in Table 9, if $L_{NLL}$ individually has already obtained most gains, I am afraid whether the claim is valid.
>
> A4:  Adding $L_{NLL}$ to $L_{DPO}$  **is a standard step in DPO-based alignment optimization**. Just like we must add Layer Normalization after the multi-head self-attention layer of the Transformer, we also must add the $L_{NLL}$ to $L_{DPO}$. We just follow previous works to add the $L_{NLL}$ to improve the stability of DPO training. For example, **the authors of LLAMA-3 also add the $L_{NLL}$ to their DPO loss**, see page 16 of the LLAMA-3 tech report [1].
>
> **We assume you are junior in this area and lack the essential background in understanding the DPO algorithm**, so you are not aware that this is a standard step to DPO, thus ask the same question over and over again. But it doesn't matter, we are glad to explain why adding the $L_{NLL}$ is a standard step for you.
>
> - Why do we add the $L_{NLL}$ term?
>
> As we mentioned in our first version (line 354), the $L_{NLL}$ term is used for **improving the stability of DPO training**. In Eq. (3), the DPO loss consists of $x_{1} = \frac{\pi_{\theta}(y_{w}|x)}{\pi_{\text{ref}}(y_{w}|x)}$ and $x_{2} = \frac{\pi_{\theta}(y_{l}|x)}{\pi_{\text{ref}}(y_{l}|x)}$. Ideally, we want $x_{1}$ to have a large value for chosen data, and $x_{2}$ to have a small value for rejected data. However, due to the theoretical limitations of DPO [2], the gradient $\frac{\partial{L_{DPO}}}{x1}$ is smaller than $\frac{\partial{L_{DPO}}}{x2}$, which means the DPO optimization towards decreasing the value of $x_{2}$ is easier than increasing the value of $x_{1}$. This is terrible because it means that the model is not learning from the positive examples, but is just overly penalizing the negative examples. So, the solution is to add the $L_{NLL}$ term to increase the likelihood of the chosen data. We want to emphasize again that **adding $L_{NLL}$ is standard practice for improving the stability of DPO training**. Just like the gradient will explode if Layer Normalization is not added to the Transformer, current mainstream DPO algorithms all mention that they have added $L_{NLL}$, such as LLAMA 3 [1].
>
> [1] The Llama 3 Herd of Models, Meta AI.
>
> [2] Towards Analyzing and Understanding the Limitations of DPO: A Theoretical Perspective
>
> - Can we only use the $L_{NLL}$ without the $L_{DPO}$?
>
> **Which do you think is more important: improving on multi-image datasets while preserving the capabilities of single-image benchmarks, or improving only on multi-image datasets and significantly decreasing on a single image?**  As clearly shown in Table 10, using only NLL loss results in a significant decrease in the model's single-image performance (LLaVA 51.6 → LLAVA+NLL 49.9, a -1.7% drop). This directly refutes your argument that `NLL loss alone accounts for the majority of performance improvements`.
>
> In contrast, our DPO method, compared to SFT, leverages negative data and the KL divergence term in the policy model (Eq. 3), which not only leads to greater improvements in multi-image understanding (LLaVA +3.0%, IXC2.5 +4.3%) but also preserves the original single-image reasoning ability (LLaVa 51.6, MIA-DPO 51.7, a +0.1% increase). These are significant gains that the NLL-only approach simply can’t achieve.
>
> While NLL loss may improve performance on multi-image benchmarks to some degree, the magnitude of the improvement and the preservation of single-image ability are both far superior in our approach. The evidence is in the numbers, and the results speak for themselves. We encourage you to carefully review the data presented in our previous response, as it clearly highlights the limitations of NLL loss and the advantages of our proposed method.
>
> We are glad to answer your further questions to address your concerns! And again, we want to ask you the following questions:
>
> - **Do you think it is reasonable to weaken the contribution of a paper by using a tech report that has not been peer-reviewed and is released in the same period?**
>
> - **Should we consider the average results on the 8 datasets, or just look at the decreasing result on one dataset and ignore the other seven?**
>
> - **Can we weaken the contribution of one paper simply because it may be optimized further in the future (but not verified)?**
>
> - **Which do you think is more important: improving on multi-image datasets while preserving the capabilities of single-image benchmarks, or improving only on multi-image datasets and significantly decreasing on a single image?**
>
> - **Many existing works in the NLP/LLM area have demonstrated that applying RLHF (e.g., using the DPO loss) is better than using the SFT (NLL loss) only, and our observations on multi-image understanding also verify this point. Why do you still think merely using NLL loss is enough?**

---

> > ### Author Response · Authors · 2024-12-03
> >
> > Dear Reviewer ePWn,
> >
> > We are sorry that we have not received a response from you since we last response to your questions. Do you have any further concerns? We will do our utmost to address them. Given the mixed results with two ratings of 6 and one of 5, we would be grateful if you could clarify any remaining concerns that we may not have fully addressed. We truly look forward to your response. Thank you again for your time and thoughtful feedback.
> >
> > With sincere appreciation,
> >
> > The Authors

---

### Author Response · Authors · 2024-11-21
**Rebuttal by Authors**

Dear Reviewers,

Thank you for taking the time to review our manuscript and for providing thoughtful and constructive feedback. We are delighted to see that the reviewer acknowledged the paper is clearly written (ePWn), clear improvements and thorough ablations (mzdh), efficient pipeline (ZLYy), and clear motivation(DZHj). Below we summarize some major points we have addressed in the rebuttal. More detailed responses are provided individually for each reviewer.

**Image Coherence.** We analyzed the limitations of MIA-DPO and also conducted experiments to demonstrate that MIA-DPO can effectively handle the primary challenges of multi-image understanding. Furthermore, it achieves performance improvements on benchmarks that require understanding multiple related images simultaneously. (Reviewer ePWn)

**SFT Baseline(only the NLL term).** We added a baseline that directly uses the chosen data from MIA-DPO for SFT and compared it with other multi-image SFT methods. These comparisons validate the effectiveness of our approach. (Reviewer ePWn, ZLYy)

**Further Details on Calculating Attention Ratio.** To address the reviewer's concerns, we conducted a more detailed analysis of the formulas presented in the paper regarding the calculation of attention ratios.  (Reviewer ePWn, DZHj)

**Impact of Data Types and Ratios.** We further analyzed the impact of different data proportions, as well as the threshold settings and the results of the ablation experiments on thresholds.(Reviewer mzdh)

**Additional Experiments.** We supplemented our work with ablation experiments on model size to explore the performance of larger models. Furthermore, we added baselines for POVID and HA-DPO on the IXC2.5 model for comprehensive comparisons. (Reviewer mzdh, DZHj)

**VQA Test Set.** We constructed a 500-question multi-image VQA test set to evaluate model performance. Additionally, we created a subset consisting of 50 questions each for 4, 6, 8, and 10 images to explore the model's capabilities when handling a larger number of images. (Reviewer ZLYy)

In the point-to-point responses below, we have provided more detailed responses to each of the reviewers' concerns. ***All new material added to the revised manuscript has been highlighted in red text for better visibility.*** Please don't hesitate to let us know if there are any additional clarifications or experiments that we can offer！

Yours Sincerely,

Authors

---

### Author Response · Authors · 2024-11-29

To All Reviewers:

I hope this message finds you well. We deeply appreciate the time and effort all reviewers have dedicated to our paper. However, we are embarrassed to say that we have not received any responses since the rebuttal period began, hence this reminder.

According to the ICLR 2025 Reviewer Guide, `The discussion phase at ICLR is different from most conferences in the AI/ML community. During this phase, reviewers, authors and area chairs engage in asynchronous discussion and authors are allowed to revise their submissions to address concerns that arise. It is crucial that you are actively engaged during this phase. Maintain a spirit of openness to changing your initial recommendation (either to a more positive or more negative) rating.`.

This year, the ICLR organization community has made significant efforts to improve review quality. We sincerely thank PCs and ACs for sending reminder messages to reviewers who have not posted their responses. We sincerely thank the PCs that decided to extend the discussion period to October 2nd to facilitate more in-depth author-reviewer discussions. Several authors of this paper also served as reviewers. We diligently read rebuttals from other papers, ensuring that concerns were addressed and asking follow-up questions as needed.

We consider accepting a review invitation to be an honor that entails a responsibility to help the authors revise their manuscript.

We believe that as a reviewer, it is crucial to respond to rebuttal, regardless of whether you think our submission should be accepted or rejected. Even if our rebuttal does not alter your initial rating, it is important to communicate that you have carefully considered our comments.

We believe that even if you are very busy or on holiday, a short acknowledgment is at least necessary and will not take up too much of your time. In fact, being too busy to reply is not a reasonable excuse. During the around three weeks of discussion, we believe that PCs and ACs are often even busier than most reviewers. Despite this, they have made significant efforts to maintain the quality of author-reviewer discussions. We think all authors and reviewers should collaborate to be involved in the discussion period for our reputation and the benefit of the ICLR community.

We are committed to participating in the discussion period. **The discussion period deadline is approaching. We kindly want to follow up and ensure the reviewers provide your feedback.**

Regard,

Authors

---

### Meta-Review · Area_Chair_vmiN · 2024-12-19

**Metareview:**

The paper studies the problem of improving VLMs for image recognition. The authors propose to construct tasks where inspecting the model's attention on a sequence of images can be used as a signal to improve the model. The authors collect a preference dataset based on the synthetic data and then use DPO to improve the model. The resulting method is evaluated using models such as LLava and Intern-LM on standard image understanding benchmarks where it improves performance.

Strengths
1. The paper proposes a simple idea of concatenating images into a sequence and use this multi-image task to get the VLMs to focus on the right image. The simplicity of this idea is a feature.
2. The experiments in this paper are well thought out and show that this work improves performance across multiple models and benchmarks.
3. The proposed method does not require creating a fresh dataset and shows that simple preference data collection can also improve performance.

Weaknesses
1. The method requires tuning the attention thresholds depending on the number of images. This parameter also seems to affect the final performance and makes the method brittle.
2. Studying the effect of multi-image DPO on larger models would be an interesting future direction since the effects of prior augmentation based training techniques typically diminishes with model size.


Justification
The paper proposes a simple, practically useful idea that is evaluated across multiple models and datasets. The primary concerns raised by the reviewers have been addressed by the authors.

**Additional Comments On Reviewer Discussion:**

The reviewers raised concerns around
1. Novelty of the work
2. Clarifications about experimental setup
3. Relation to unpublished work
4. Using larger models
5. Simplicity of the work

Overall, the authors addressed the concerns and the AC feels that the major remaining concern around novelty/simplicity isn't well founded and grounds for rejection of this work.

---

### Decision · Program_Chairs · 2025-01-22

Accept (Poster)